# Condensin minimizes topoisomerase II-mediated entanglements of DNA *in vivo*

Sílvia Dyson, Joana Segura, Belén Martínez-García, Antonio Valdés & Joaquim Roca[*]

## Abstract

The juxtaposition of intracellular DNA segments, together with the DNA-passage activity of topoisomerase II, leads to the formation of DNA knots and interlinks, which jeopardize chromatin structure and gene expression. Recent studies in budding yeast have shown that some mechanism minimizes the knotting probability of intracellular DNA. Here, we tested whether this is achieved via the intrinsic capacity of topoisomerase II for simplifying the equilibrium topology of DNA; or whether it is mediated by SMC (structural maintenance of chromosomes) protein complexes like condensin or cohesin, whose capacity to extrude DNA loops could enforce dissolution of DNA knots by topoisomerase II. We show that the low knotting probability of DNA does not depend on the simplification capacity of topoisomerase II nor on the activities of cohesin or Smc5/6 complexes. However, inactivation of condensin increases the occurrence of DNA knots throughout the cell cycle. These results suggest an *in vivo* role for the DNA loop extrusion activity of condensin and may explain why condensin disruption produces a variety of alterations in interphase chromatin, in addition to persistent sister chromatid interlinks in mitotic chromatin.

**Keywords** chromatin; DNA knot; DNA loop extrusion; DNA topology; SMC complex
**Subject Categories** Chromatin, Transcription & Genomics; DNA Replication, Recombination & Repair
**The EMBO Journal (2021) 40: e105393**

## Introduction

Type-2A topoisomerases, such as bacterial topo IV and eukaryotic topo II, pass one segment of duplex DNA through the transient double-stranded DNA break that they produce in another segment (Wang, 1998). This DNA-passage activity is essential to remove the intertwines generated between newly replicated DNA molecules and to modulate DNA supercoiling during genome transactions (Corbett & Berger, 2004; Nitiss, 2009). However, the activity of type-2A topoisomerases also entails important threats. Firstly, the DNA cleaving step can be a source of chromosomal damage (Nitiss &

Wang, 1996). Secondly, the DNA-passage activity can entangle DNA molecules that are closely packed or folded via protein-DNA interactions (Hsieh, 1983; Wasserman & Cozzarelli, 1991; Roca *et al*, 1993). Indeed, computer simulations have predicted that DNA molecules confined in biological systems would be massively entangled if type-2A topoisomerases could freely equilibrate their global topology (Arsuaga *et al*, 2002; Micheletti *et al*, 2008; Dorier & Stasiak, 2009). Fortunately, this prospect does not occur because the hierarchical folding of chromatin, which has a scaling behavior similar to that of a fractal globule, drastically reduces the topological complexity of chromosomes (Lieberman-Aiden *et al*, 2009; Mirny, 2011). Accordingly, 3D analyses of the eukaryotic nuclear architecture revealed little intermingling of chromosomal territories and large chromatin domains (Denker & de Laat, 2016; Schmitt *et al*, 2016). Reconstruction of 3D paths of high-order chromatin fibers in individual cells also evidenced the scarcity of long-range entanglements (Siebert et al., 2017, Stevens *et al*, 2017; Sulkowska *et al*, 2018).

The hierarchical architecture of chromatin, however, cannot prevent the formation of DNA interlinks between chromatin fibers that come in close proximity or the formation of DNA knots within clusters of nucleosomes. Another mechanism must thereby operate to avoid these local DNA entanglements. This is exemplified by the sister chromatid interlinks (SCI), which are eliminated during prophase, even though sister chromatids remain cohesed until metaphase (Nagasaka *et al*, 2016). Further evidence of such mechanism also emerged from the analysis of the knotting probability ($P^{kn}$) of intracellular chromatin (Valdes *et al*, 2018). These studies revealed that topo II-mediated knotting and unknotting of DNA normally occur within stretches of 10 to 60 nucleosomes (Fig 1A). However, the $P^{kn}$ of chromatin does not scale proportionally to DNA length, as would be expected for any polymer chain. The slope of $P^{kn}$ is progressively reduced in domains larger than 20 nucleosomes (Valdes *et al*, 2018), as if some mechanism were counteracting the potential entanglement of intracellular DNA (Fig 1B).

Two mechanisms have been hypothesized that could minimize the entanglement of DNA *in vivo*. The first one relies on the intrinsic capacity of type-2A topoisomerases to simplify the equilibrium topology of DNA molecules in free solution (Rybenkov *et al*, 1997). Namely, topo II uses a three-gate mechanism to pass one segment of DNA (T-segment) through another (G-segment) in an ATP-dependent manner (Wang, 1998). Upon topo II binding to the G-segment,

---

Molecular Biology Institute of Barcelona (IBMB), Spanish National Research Council (CSIC), Barcelona, Spain
*Corresponding author. Tel: +34 93 4020117; E-mail: joaquim.roca@ibmb.csic.es

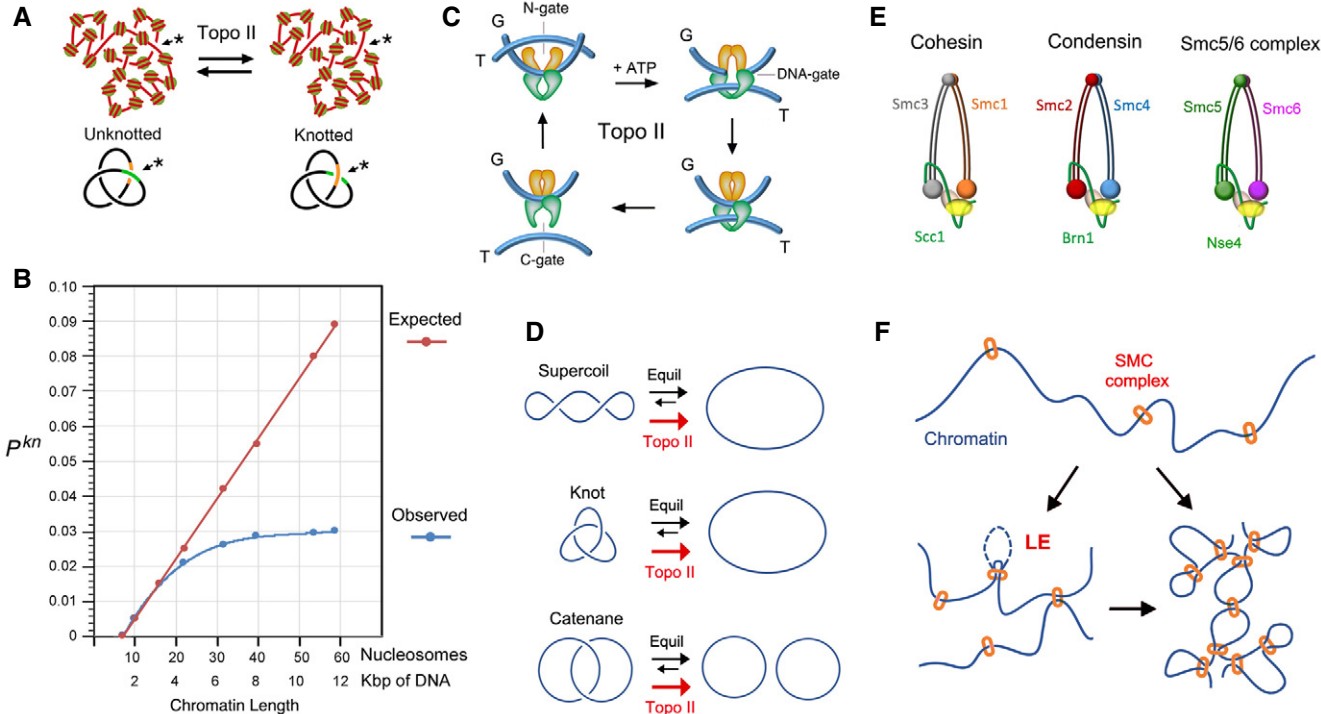

**Figure 1. Knotting probability of intracellular DNA and plausible regulatory mechanisms.**

A  Topo II activity on random juxtapositions of DNA segments (*) produces steady-state fractions of DNA knots in intracellular chromatin.

B  DNA knotting probability ($P^{kn}$) of intracellular chromatin (observed) does not scale proportionally to the length of DNA (expected). The slope of $P^{kn}$ is reduced in chromatin stretches larger than 20 nucleosomes. Data from (Valdes et al., 2018).

C  Three-gate mechanism of topo II to pass one segment of DNA (T-segment) through another (G-segment). Upon ATP binding, the T-segment is captured by the entrance gate (N-gate) and passed through the transiently cleaved G-segment (DNA-gate). Upon re-ligation of the G-segment, the T-segment is released through the exit gate (C-gate).

D  Topo II activity reduces the fractions of DNA supercoils, knots and catenates to below the topological equilibrium values (see details in Fig EV1).

E  Architecture of the SMC complexes of *S. cerevisiae*. The Smc heterodimers (Smc1-Smc3, Smc2-Smc4, Smc5-Smc6) and kleisin (Scc1, Brn1, Nse4) subunits of cohesin, condensin, and the Smc5/6 complex are indicated.

F  SMC complexes entrap segments of DNA to form chromatin loops and/or bridge nearby chromatin domains. Their loop extrusion activity (LE) ensures the co-entrapment of contiguously oriented intramolecular DNA segments.

the T-segment is captured by closing the entrance gate (N-gate) of the enzyme. The T-segment is then passed through the transiently cleaved G-segment (DNA-gate), and it is released outside the enzyme through the exit gate (C-gate) (Fig 1C). This mechanism allows DNA supercoils, knots, and catenates to be reduced. However, when topo II relaxes supercoiled DNA, it produces a linking number (Lk) distribution of DNA topoisomers that is narrower than the equilibrium Lk distribution (Figs 1D and EV1A). Likewise, when topo II unknots or decatenates DNA molecules, it reduces the fraction of knotted and catenated molecules to values below the topological equilibrium (Figs 1D and EV1B). The mechanism by which topo II is able to assess and locally reduce the equilibrium topology of large DNA molecules remains mysterious (Vologodskii, 2016). Moreover, the physiological relevance of this simplification activity is unknown since it has never been assessed *in vivo*. Yet, *in vitro* studies have shown that topo II does not simplify the equilibrium topology of DNA when the enzyme activity is quenched with the inhibitor ICRF-193 (Fig EV2A) or when the C-gate of the enzyme is deleted (Fig EV2B) (Martinez-Garcia *et al*, 2014; Thomson *et al*, 2014). These two observations opened up the

possibility to target the simplification activity of intracellular topo II and test whether that affects the $P^{kn}$ of chromatin.

The second mechanism that could reduce intracellular DNA entanglements relies on the activity of structural maintenance of chromosomes (SMC) complexes, which are mainly identified as cohesin, condensin, and the Smc5/6 complex in eukaryotic cells (Uhlmann, 2016; Yatskevich *et al*, 2019). Cohesin generates the DNA loops that organize chromatin during interphase and holds sister chromatids together from S-phase until metaphase (Onn *et al*, 2008; Nasmyth & Haering, 2009). Condensin plays a key role in the compaction and individualization of chromatids during cell divisions (Hirano, 2012). The Smc5/6 complex has been mainly implicated in DNA repair via homologous recombination (Aragon, 2018). Despite their distinct roles, SMCs have similar architecture. They are large rod-shaped proteinic ensembles composed of a trimeric core formed by a heterodimer of Smc ATPases and a conserved kleisin, in addition to several additional regulatory subunits (Fig 1E). ATP binding and hydrolysis produce the opening and closure of distinct SMC compartments, which can embrace one or more segments of DNA (Hassler *et al*, 2018; Yatskevich *et al*, 2019).

Moreover, ATP usage can produce the translocation of the SMC complex along DNA (Terakawa *et al*, 2017) and the extrusion of DNA loops (Ganji *et al*, 2018; Davidson *et al*, 2019; Kim *et al*, 2019) (Fig 1F). The notion that SMCs can promote the removal of DNA entanglements was proposed in the context of sister chromatid resolution (Sen *et al*, 2016; Piskadlo *et al*, 2017). Former studies suggested that positive supercoils generated by condensin in mitotic chromosomes could produce a bias in topo II activity to eliminate the SCIs (Baxter *et al*, 2011; Sen *et al*, 2016). Subsequent computational simulations showed that DNA loop extrusion activity of SMCs would constrict DNA entanglements and so bias topo II to disentangle intermixed chromosomes (Goloborodko *et al*, 2016a) and minimize the occurrence of DNA knots (Racko *et al*, 2018; Orlandini *et al*, 2019).

Here, we show that precluding the capacity of topo II to simplify equilibrium topology of DNA does not alter the low knotting probability of intracellular chromatin. Inactivation of cohesin or the smc5/6 complex also does not increase knot formation. However, inactivation of condensin markedly increases the occurrence of chromatin knots throughout the cell cycle. We propose that the requirement of condensin to minimize DNA entanglements might rely on its DNA loop extrusion activity. This function could explain the wide range of alterations that condensin inactivation produces both in interphase and mitotic chromatin.

## Results

### Topoisomerase II does not minimize the knotting probability of chromatin

We used two experimental approaches to assess whether the capacity of topo II to simplify the equilibrium topology of DNA was sustaining the low knotting probability ($P^{kn}$) of intracellular chromatin. First, we used the topo II inhibitor ICRF-193 to impair the simplification activity of cellular topo II (Fig EV2A) (Martinez-Garcia *et al*, 2014). We carried out this experiment in a $\Delta top1$ *TOP2* yeast strain that hosted the circular minichromosome YEp13 (10.7 Kb) as the reporter of $P^{kn}$ (Fig 2A). To verify that the simplification capacity of yeast topo II was targeted by ICRF-193, we added to crude lysates of the cells a negatively supercoiled DNA plasmid (YEp24, 7.8 Kb), which served as internal control of topo II activity (Fig 2A). When YEp24 was relaxed by a purified type-1B topoisomerase (topo I) (Fig 2B, lanes 1 and 2), it produced an equilibrium distribution of Lk topoisomers (Fig EV1A). However, when YEp24 was relaxed by the topo II activity present in the cell lysates, it produced a distribution of Lk topoisomers that was narrower than the equilibrium Lk distribution generated by topo I (Fig 2B, compare Lk plots of lanes 2 and 3). When we quenched the topo II activity by adding ICRF-193 to the mixture, the Lk distribution of YEp24 became broadened to an extent similar to that of the equilibrium Lk distribution (Fig 2B, compare Lk plots of lanes 2 and 4). Therefore, cellular topo II was able to simplify the equilibrium topology of naked DNA and this capacity was canceled by ICRF-193. We then examined what happened to the topology of the YEp13 minichromosome present in the above mixtures before and after the addition of ICRF-193 (Fig 2C). To this end, we first ran a 2D-gel electrophoresis containing chloroquine (Hanai & Roca, 1999) to resolve the Lk

distribution of the YEp13 DNA (Fig EV3). Before adding ICRF-193, YEp13 presented a distribution of topoisomers of negative $\Delta Lk$ values (Fig 2C, Lk) consistent with the negative supercoils constrained by nucleosomes (Segura *et al*, 2018) (Fig EV3). Following the addition of ICRF-193, the Lk distribution of YEp13 was not significantly altered, which contrasted to what was observed in the control plasmid YEp24 (compare Lk plots in Fig 2C and 2B). Next, we nicked the DNA samples and ran a different 2D gel electrophoresis to reveal DNA knots (Trigueros *et al*, 2001)(Fig EV4). Both before and after adding ICRF-193, YEp13 presented a similar ladder of knotted molecules (Fig 2C, Kn), which started with the knot of three irreducible crossings (trefoil knot or $3_1$) as the most abundant form (Fig EV4). We calculated $P^{Kn}$ as the relative abundance of total knotted molecules with respect to the total amount of unknotted and knotted DNA circles. In agreement with previous studies, the $P^{kn}$ of YEp13 was about 0.03 (Valdes *et al*, 2018), three times lower than the value expected if $P^{kn}$ scaled proportionally to DNA length (Fig 1B). Following the addition of ICRF-193, the $P^{kn}$ of YEp13 was not significantly altered (Fig 2C).

Our second approach to test the simplification activity of topo II on chromatin was via the expression of a truncated topo II (*Top2-Δ83*), in which the C-gate was removed (Martinez-Garcia *et al*, 2014) (Fig 2D, Fig EV2B). Similar to the type-2B class of topoisomerases that innately lack a C-gate (Fig EV2C), *Top2-Δ83* is able to reduce DNA topology constraints but cannot simplify the equilibrium topology of free DNA (Martinez-Garcia *et al*, 2014; Thomson *et al*, 2014). We used the galactose-inducible *pGal1* promoter to express in $\Delta top1$ *top2-4* yeast cells either *Top2-Δ83* or *TOP2* (full-length topo II) enzymes (Fig 2D). Upon inactivation of the *top2-4* thermo-sensitive allele, we examined the effects of the expressed enzymes on the topology of the control plasmid YEp24 and the minichromosome Yep13 present in these cells. As expected, in cells expressing *TOP2*, YEp24 was relaxed and presented a narrow (i.e., simplified) distribution of Lk topoisomers, whereas in cells expressing *Top2-Δ83*, the resulting Lk distribution was wider (Fig 2E). However, neither the Lk distribution nor the knotting probability of the Yep13 minichromosome was altered in the presence of the *TOP2* and *Top2-Δ83* activities (Fig 2F). Therefore, in concordance with the ICRF-193 results, precluding the simplification capacity of topo II did not produce any significant effect on the topology of chromatinized DNA.

### Condensin inactivation boosts the occurrence of chromatin knots

To test whether the low knotting probability of intracellular DNA was achieved via the activity of SMC complexes, we examined the DNA topology of the Yep13 minichromosome in yeast strains previously characterized for carrying thermo-sensitive mutations that inactivate either condensin (*smc2-8*) (Freeman *et al*, 2000), cohesin (*scc1–73*)(Michaelis *et al*, 1997), or the Smc5/6 complex (*smc6-9*) (Torres-Rosell *et al*, 2005) (Appendix Fig S2). In each case, we grew the cells at a permissive temperature (26°C) and, upon reaching the exponential phase (OD 0.6-0.8), we shifted one half of the cultures to 35°C for 60 min. We then fixed the topology of intracellular DNA by quenching the cells with a cold ethanol-toluene solution and extracted their total DNA (Diaz-Ingelmo *et al*, 2015). As in the foregoing experiments, we ran a 2D-gel electrophoresis to examine the distribution of Lk topoisomers of YEp13; and we nicked the DNA to

examine the occurrence of DNA knots in a different 2D gel electrophoresis.

The Lk distribution of YEp13 in the three strains presented negative ΔLk values, which were not significantly altered upon inactivation of condensin, cohesin, or the Smc5/6 complex (Fig 3A-C).

Likewise, before inactivation of the SMCs, the knotting probability of YEp13 was low and similar in the three strains ($P^{kn} \approx 0.03$) (Fig 3A-C). This concordance indicated that the knot minimization mechanism is robust and performs equally in most cells. However, upon inactivation of condensin, $P^{kn}$ of YEp13 increased about

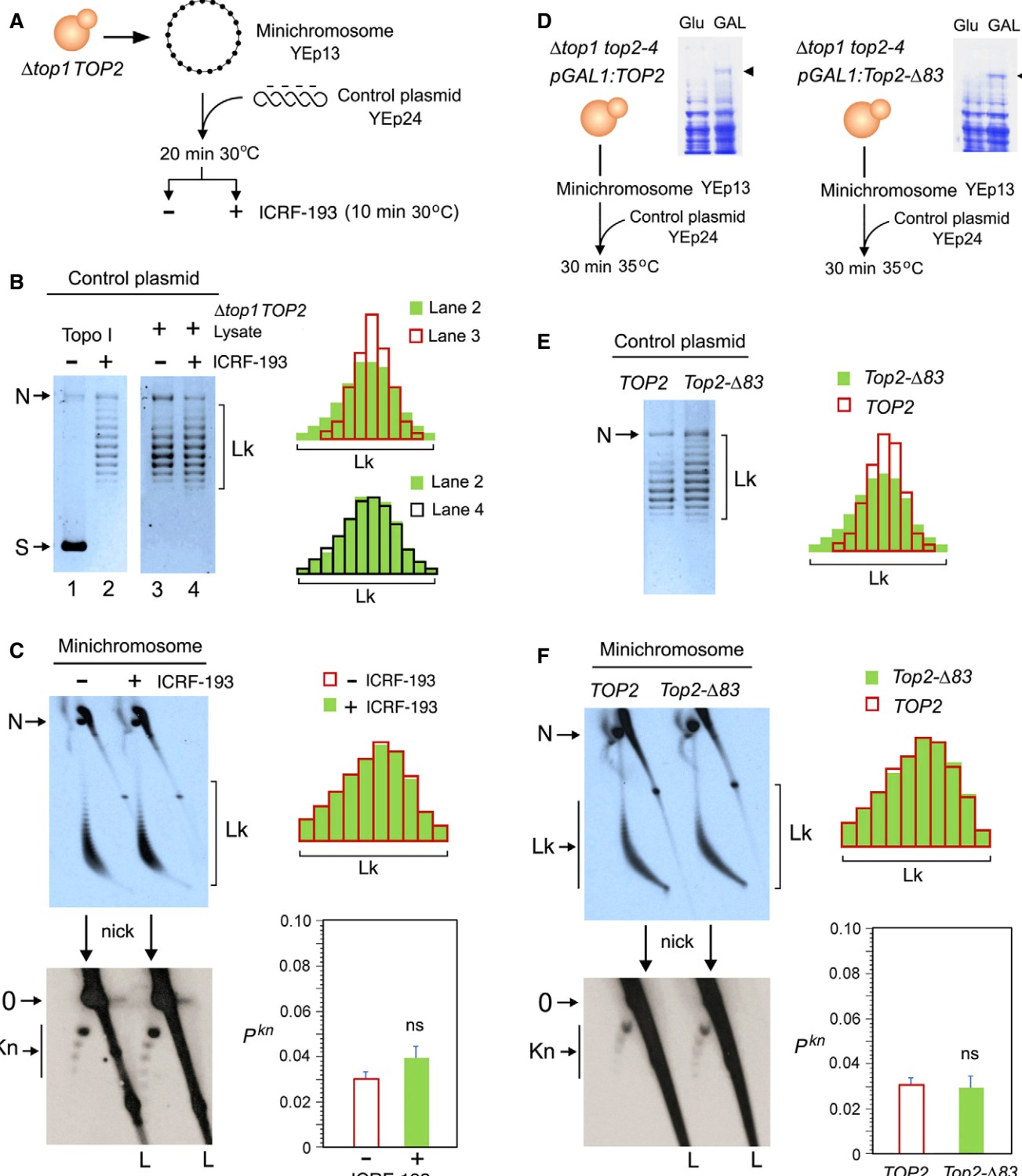

Figure 2.

**Figure 2.  Topoisomerase II does not minimize the knotting probability of chromatin.**

A   Experimental layout to test the DNA topology simplification activity of cellular topo II upon the addition of ICRF-193.

B   Lanes 1 and 2: negatively supercoiled plasmid (YEp24) and its equilibrium distribution of Lk topoisomers upon its relaxation with Topo I. Lanes 3 and 4: distribution of Lk topoisomers of YEp24 upon its relaxation in lysates of Δ*top1 TOP2* yeast cells in absence and after the addition of ICRF-193. Plots compare the relative intensity of individual topoisomers of the Lk distributions in lanes 2, 3, and 4.

C   Top: 2D gel electrophoresis of the Lk distributions of the YEp13 minichromosome present in the lysates of Δ*top1 TOP2* yeast cells before and after the addition of ICRF-193 (see details in Fig EV3). Plots compare the relative intensity of the Lk distributions (divided into ten sections). Bottom: 2D gel electrophoresis of the same samples upon nicking the DNA in order to reveal the occurrence of knots (see details in Fig EV4). The graph shows $P^{kn}$ of YEp13 (mean ± SD from three independent experiments). P-values (Student's t test): ns, $P > 0.05$.

D   Experimental layout to compare the activities of *TOP2* and *Top2-Δ83* on DNA and chromatin. Arrowheads indicate the extrachromosomal expression of *TOP2* and *Top2-Δ83* under the inducible *pGAL1* promoter.

E   Lk distributions of the control plasmid (YEp24) relaxed by lysates of Δ*top1 top2-4* yeast cells that expressed *TOP2* or *Top2-Δ83*. Plots compare the relative intensity of individual Lk topoisomers.

F   Top: 2D gel electrophoresis of the Lk distributions of the YEp13 minichromosome produced in the presence of *TOP2* or *Top2-Δ83*. Plots compare the relative intensity of the Lk distributions (divided into ten sections). Bottom: 2D gel electrophoresis of the same samples upon nicking the DNA in order to reveal the occurrence of knots. Graph: $P^{kn}$ of YEp13 (mean ± SD from three independent experiments).

Gel signals are: N, nicked DNA circles; S, supercoiled DNA; L, diagonal of linear DNA fragments; Lk, distribution of Lk topoisomers; 0, unknotted DNA (nicked); Kn, ladder of knotted forms (nicked). P-values (Student's t test): ns, $P > 0.05$.

threefold ($P^{kn} \approx 0.09$) (Fig 3A). Inactivation of cohesin produced a slight yet not significant reduction ($P^{kn} \approx 0.02$) (Fig 3B). Inactivation of the Smc5/6 complex did not change the knot abundance (Fig 3C). To verify that the *smc2-8* allele was causing the threefold increase of $P^{kn}$, we introduced this mutation in strains JCW25 (*TOP2*) and JCW26 (*top2-4*) (Trigueros & Roca, 2001). Upon shifting these cells to 35°C, the $P^{kn}$ of YEp13 increased again about threefold in the *smc2-8 TOP2* cells (Fig EV5). However, DNA knot formation did not change in the *smc2-8 top2-4* double mutant, which corroborated that topo II activity is required to produce the $P^{kn}$ changes induced by condensin (Fig EV5).

Since the above experiments were done in asynchronous cell cultures, we considered whether the effects of SMC inactivation on $P^{kn}$ would occur at different stages of the cell cycle. We conducted analogous experiments in cells arrested in $G_1$ and in metaphase (Fig 3D-3I). Arrested cells were sampled at 26°C and after shifting them to 35°C for 60 min during the arrest. Prior inactivation of the SMCs, $P^{kn}$ of YEp13 in the $G_1$ and the metaphase-arrested cells were similar to that observed in the asynchronous cell cultures ($P^{kn} \approx 0.03$) (Fig 3D-3I). This observation corroborated previous indications that the knotting probability of intracellular chromatin is not cell cycle-dependent (Valdes *et al*, 2018). Upon inactivation of condensin, the occurrence of knots in YEp13 increased about threefold both in $G_1$ ($P^{kn} \approx 0.10$) and in metaphase-arrested cells ($P^{kn} \approx 0.09$) (Fig 3D and 3E). Inactivation of cohesin produced a slight reduction of $P^{kn}$ in $G_1$ ($P^{kn} \approx 0.017$) and metaphase cells ($P^{kn} \approx 0.022$) (Fig 3F and 3G). Inactivation of the Smc5/6 complex did not alter the knot abundance at any stage (Fig 3H and 3I). Thus, we concluded that inactivation of condensin markedly increases the occurrence of DNA knots throughout the cell cycle. Remarkably, this change of $P^{kn}$ occurred without any notable alteration of the Lk distribution of the minichromosome. Therefore, the regulation of $P^{kn}$ by condensin did not involve changes of DNA supercoiling or a major disruption of chromatin structure.

### Condensin inactivation restores the DNA length-dependent entanglement of chromatin

Next, we asked whether the effects of condensin, cohesin, and Smc5/6 activity on the knotting probability of YEp13 were reproduced in other chromatin constructs. To this end, we transformed the SMCs mutant strains with circular minichromosomes that contained distinct functional elements (replication origins, transcription units, centromeres) and differed in DNA length. We inspected the topology of minichromosomes YRp3 (3.2 kb), YRp4 (4.4 kb), YRp5 (5.0 kb), YCp50 (7.9 kb), YRp21 (11.7 Kb), and of the endogenous 2-micron plasmid (6.3 kb) present in yeast cells (Appendix Fig S1). As in the foregoing experiments, we sampled exponentially growing cultures before and after inactivation of the SMCs.

In the *smc2-8* mutant (Fig 4), condensin inactivation did not significantly change the knot probability of YRp3, YRp4, and the 2-micron plasmid (Fig 4A–C). However, it augmented the occurrence of knots about twofold in YCp50 (Fig 4D), and over threefold in YRp21 (Fig 4E). Therefore, the effect of condensin inactivation on $P^{kn}$ values appeared to vary with DNA length rather than with the presence of specific functional elements. Furthermore, plotting the $P^{kn}$ changes of the distinct minichromosomes revealed that the inactivation of condensin increased $P^{kn}$ to the levels expected if knot formation was to escalate proportionally to DNA length (Fig 4F).

In the *scc1–73* mutant (Fig 5), YRp3 and YRp4 could not be analyzed since the strain was *TRP+*. Cohesin inactivation did not change the knot probability of YRp5 and the 2-micron plasmid (Fig 5A and B). However, similar to that observed in YEp13 (Fig 3B), cohesin inactivation produced a slight reduction of $P^{kn}$ in YCp50 and YRp21 (Fig 5C and D). Plotting these $P^{kn}$ values versus the size of the minichromosomes revealed that the reduction of knot formation observed in the large minichromosomes (YCp50, YEp13, and YRp21) was overall significant (Fig 5E). Finally, as in the case of YEp13, inactivation of the Smc6/5 complex did not change DNA knotting probability in any of the constructs inspected (YRp4, 2-micron plasmid, YCp50 and YRp21) (Fig 6A–E).

The above experiments corroborated that the Lk distribution of the different minichromosomes did not change upon inactivation of the SMCs, thereby excluding that $P^{kn}$ changes were consequent to alterations of DNA supercoiling or chromatin structure. The above results also evidenced that, before inactivation of the SMCs, the slope of $P^{kn}$ as minichromosomes increased in size (Figs 4F, 5E, 6E) was alike in all the strains. This similarity corroborated that the knot minimization mechanism is constitutive and robust. This mechanism is apparently sustained by the activity of condensin, since its

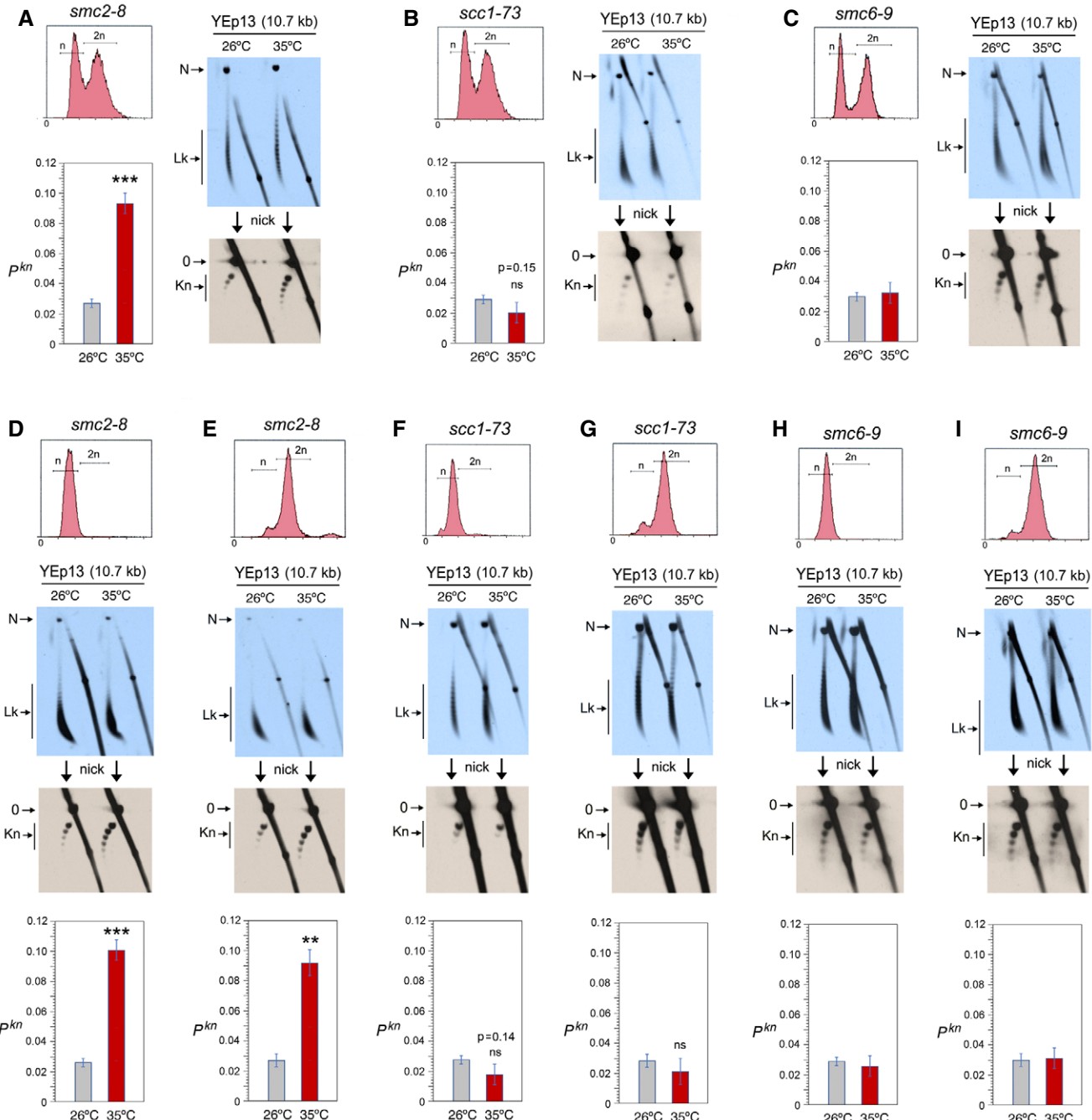

**Figure 3. Condensin inactivation boosts the occurrence of chromatin knots.**

A	Top, DNA content (n/2n) of exponentially growing ($OD_{600}$ = 0.6–0.8) *smc2-8* yeast cells. First blot: 2D gel electrophoresis of the distribution of Lk topoisomers (Lk) of the YEp13 DNA in cells quenched at 26°C and after shifting the culture to 35°C for 60 min. Second blot: 2D gel electrophoresis of the same samples upon nicking the YEp13 DNA in order to reveal the occurrence of knots (kn). Graph: $P^{kn}$ of YEp13 before and after the inactivation of condensin.

B	Experiments conducted as in (A) but in *scc1-73* yeast cells. Graph: $P^{kn}$ of YEp13 before and after the inactivation of cohesin.

C	Experiments conducted as in (A) but in *smc6-9* yeast cells. Graph: $P^{kn}$ of YEp13 before and after the inactivation of the Smc5/6 complex.

D, E	Experiments conducted as in (A), but in cells arrested in $G_1$ with alpha-factor (D) or in metaphase with nocodazole (E) for 2 h at 26°C and for one additional hour at 26°C or 35°C.

F, G	Experiments conducted as in (B), but in cells arrested in $G_1$ with alpha-factor for 2 h at 26°C (F) or in metaphase with nocodazole (G) and for one additional hour at 26°C or 35°C.

H, I	Experiments conducted as in (C), but in cells arrested in $G_1$ with alpha-factor (H) or in metaphase with nocodazole (I) for 2 h at 26°C and for one additional hour at 26°C or 35°C.

Data information: Gel signals (N, Lk, 0, Kn) are as described in Fig 2. Graphs show mean ± SD from three independent experiments in (A, B, D, E, F, G); and from two independent experiments in (C, H, I). $P$-values (Student's t test): ns, $P > 0.05$; **$P < 0.01$; ***$P < 0.001$.

	

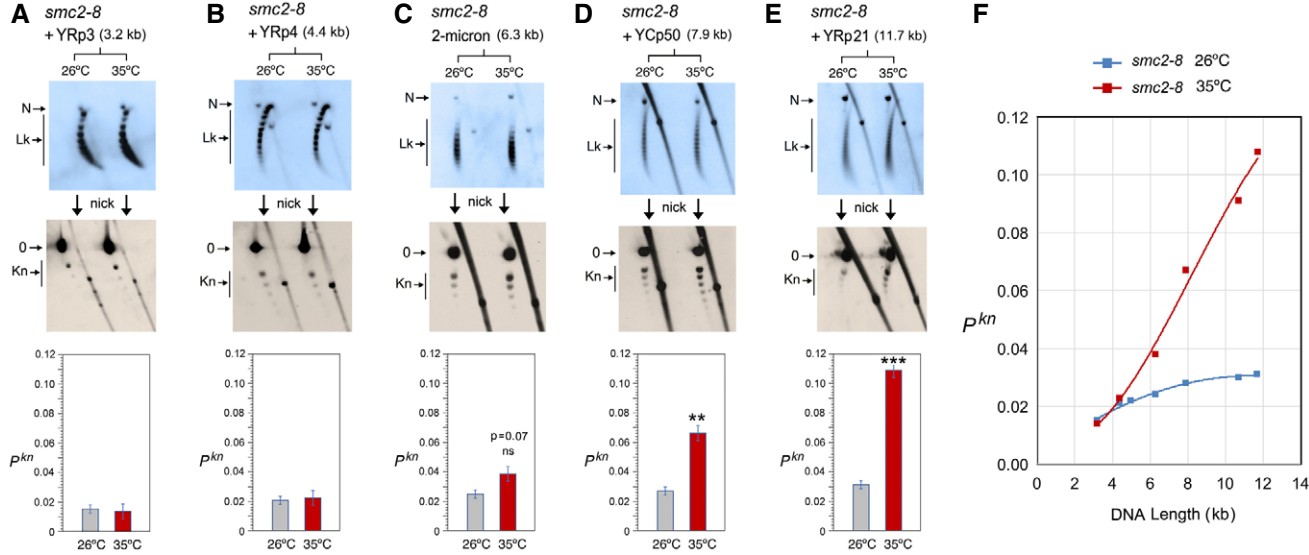

**Figure 4.   DNA length dependence of the topological effects of condensin inactivation.**

A–E   DNA topology of the indicated minichromosomes of increasing DNA length (kb) before (26°C) and after inactivation of condensin (35°C) in *smc2-8* cells. In each case, the first 2D gel resolves the Lk topoisomers (Lk), the second 2D gel uncovers the knotted forms (Kn). Gel signals are denoted as in Fig 2. Bottom graphs compare the $P^{kn}$ before and after the inactivation of condensin (mean ± SD from three independent experiments). P-values (Student's t test): ns, p> 0.05; **p < 0.01; ***p < 0.001.

F   Plot of $P^{kn}$ values of minichromosomes of increasing DNA length (including YEp13) before and after inactivation of condensin.

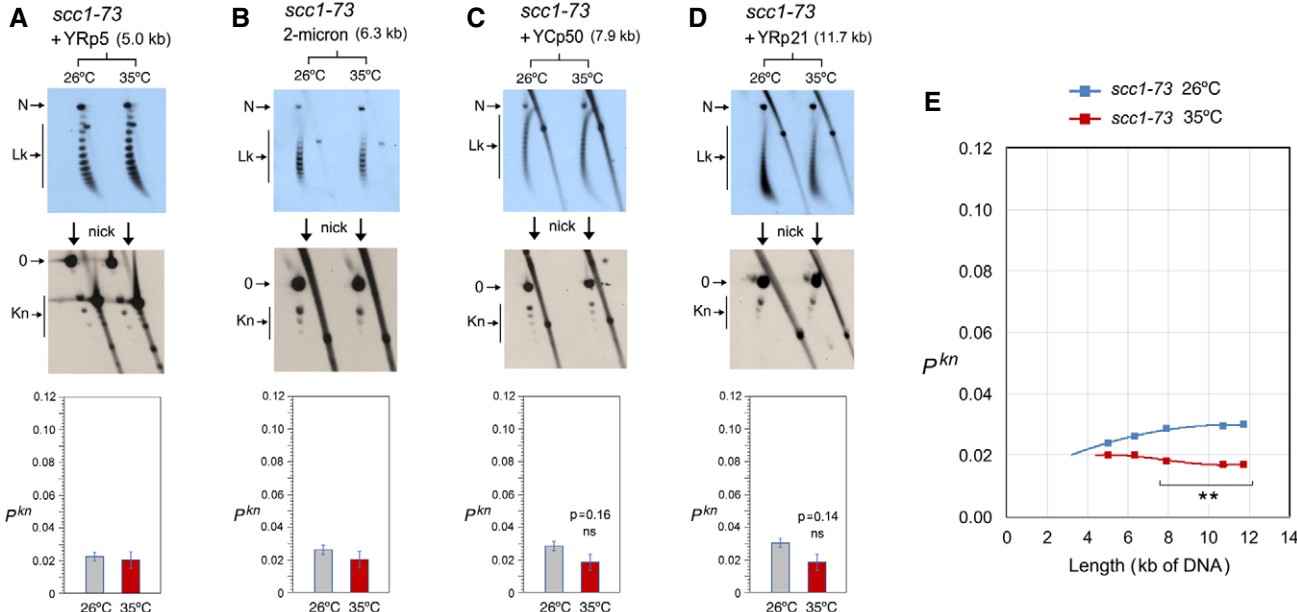

**Figure 5.   DNA length dependence of the topological effects of cohesin inactivation.**

A–D   DNA topology of the indicated minichromosomes of increasing DNA length (kb) before (26°C) and after inactivation of cohesin (35°C) in *scc1-73* cells. In each case, the first 2D gel resolves the Lk topoisomers (Lk), the second 2D gel uncovers the knotted forms (Kn). Gel signals are denoted as in Fig 2. Bottom graphs compare the $P^{kn}$ before and after the inactivation of cohesin (mean ± SD from three independent experiments). P-values (Student's *t* test): ns, P > 0.05.

E   Plot of $P^{kn}$ values of minichromosomes of increasing DNA length (including YEp13) before and after inactivation of cohesin. P-values (Student's *t* test): **P < 0.01.

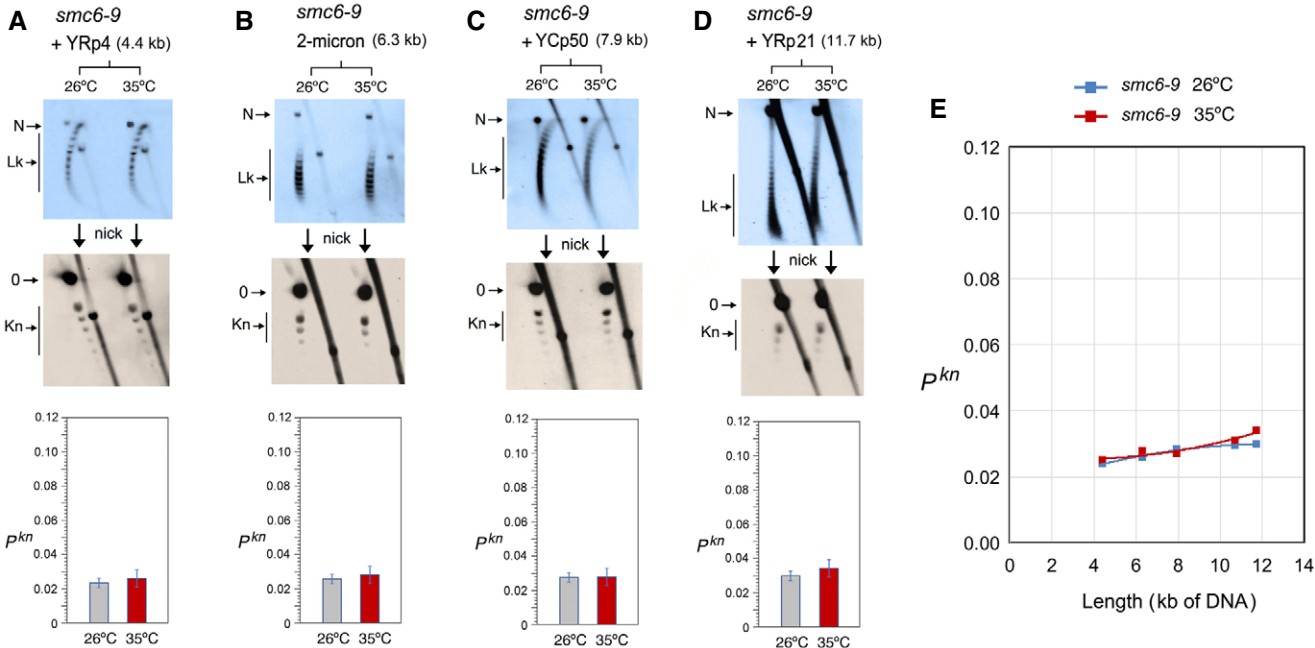

**Figure 6. DNA length dependence of the topological effects of Smc5/6 complex inactivation.**

A–D  DNA topology of the indicated minichromosomes of increasing DNA length (kb) before (26°C) and after inactivation of Smc5/6 complex (35°C) in *smc6-9* cells. In each case, the first 2D gel resolves the Lk topoisomers (Lk), and the second 2D gel uncovers the knotted forms (Kn). Gel signals are denoted as in Fig 2. Bottom graphs compare the $P^{kn}$ before and after the inactivation of Smc5/6 complex (mean ± SD from two independent experiments).

E  Plot of $P^{kn}$ values of minichromosomes of increasing DNA length (including YEp13) before and after inactivation of Smc5/6 complex.

inactivation restored the DNA length-dependent entanglement of chromatin (Fig 4F).

# Discussion

The intrinsic capacity of topoisomerase II to simplify the equilibrium topology of DNA in free solution is commonly stated as the mechanism that prevents indiscriminate entanglement of intracellular DNA. This assumption, however, had never been experimentally tested until the present study. Our results show that disrupting the simplification activity of cellular topo II does not increase DNA knotting in chromatin. Apparently, the equilibrium topology of chromatinized DNA is not recognized by topo II in the same way as in free DNA. While these negative results cannot formally discard some role of the simplification capacity of topo II *in vivo*, the marked effects produced by condensin indicate that minimizing the entanglement of intracellular DNA mainly depends on this SMC complex.

### Mechanism of condensin to minimize DNA entanglements

Our finding that condensin minimizes the knotting probability of intracellular DNA seems a priori counterintuitive. Normally, any condition that folds or compacts DNA should promote its topological entanglement, not the opposite. Consistent with this notion, early *in vitro* studies found that condensin markedly increases topo

II-mediated knotting of DNA (Kimura *et al*, 1999; Losada & Hirano, 2001). DNA knotting and catenation were also found stimulated by cohesin (Losada & Hirano, 2001), the Smc5/6 complex (Kanno *et al*, 2015), and bacterial SMCs (Petrushenko *et al*, 2006; Bahng *et al*, 2016). These observations supported the notion that SMCs can embrace or bring in close proximity two or more segments of DNA. However, since SMCs had to be added in large molar excess (> 30:1) over circular DNA molecules to stimulate knotting or catenation, these experiments did not reflect a physiological context. Conversely, current evidence that individual condensin complexes can translocate along DNA (Terakawa *et al*, 2017) and produce the extrusion of DNA loops (Ganji *et al*, 2018) explain how condensin might promote the removal of DNA knots. Computational simulations of LE activity indicated that the extrusion process would tighten any intra- or inter-molecular entanglement of DNA and enforce its removal by topo II (Goloborodko *et al*, 2016a; Racko *et al*, 2018; Orlandini *et al*, 2019). As a result, LE activity would reduce the equilibrium fractions of DNA links and knots, whereas LE inactivation would reestablish the equilibrium fractions (i.e., random entanglements of the DNA), which escalate proportionally to DNA length (Frank-Kamenetskii *et al*, 1975; Rybenkov *et al*, 1993; Shaw & Wang, 1993). Remarkably, this prospect matches with the effects of condensin inactivation on minichromosomes of increasing size (Fig 4F).

Since condensin minimizes intramolecular entanglements of DNA (knots), it might operate similarly to remove inter-molecular DNA tangles such as the sister chromatid interlinks (SCI) that arise

during DNA replication. A compaction-independent role of condensin has been involved in the removal of these linkages (D'Amours *et al*, 2004; Renshaw *et al*, 2010). Moreover, although sister chromatids remain in very close proximity by the effect of cohesin until anaphase, the removal of SCI is nearly completed at the end of prophase (Nagasaka *et al*, 2016). However, inactivation of condensin during metaphase results in de novo formation of SCI (Sen *et al*, 2016; Piskadlo *et al*, 2017), which implies that condensin promotes the unlinking of sister chromatids while their close proximity still favors interlinking. In this respect, it was proposed that positive DNA supercoils generated by condensin in mitotic chromatin produce a bias in topo II function to remove the SCIs (Baxter *et al*, 2011; Sen *et al*, 2016). However, *in vitro* studies indicated that condensin does not compact DNA by inducing DNA supercoiling (Eeftens *et al*, 2017). Moreover, recent *in vivo* studies have shown that positive supercoiling of DNA markedly increases the formation of DNA knots (Valdes *et al*, 2019). Accordingly, if condensin were generating supercoils to promote the removal of SCI, that would in turn increase knot formation in mitotic chromatin. This prospect is inconsistent with our results, which show that condensin minimizes the occurrence of knots without altering DNA supercoiling both in interphase and mitotic chromatin. Therefore, our findings support the notion that the removal of intra- and inter-molecular DNA entanglements could be promoted via the LE activity of condensin (Fig 7).

### Distinct effects of condensin and cohesin

In contrast to condensin, inactivation of cohesin and the smc5/6 complex did not increase knot formation. Moreover, cohesin inactivation slightly reduced $P^{kn}$ both in G1 and mitotic cells. This observation indicates that the plausible implication of cohesin on knot formation must be independent of its role in sister chromatid cohesion. In that case, the distinct effects of condensin and cohesin on $P^{kn}$ are striking since both complexes have LE activity (Davidson *et al*, 2019; Kim *et al*, 2019). Indeed, LE activity of cohesin *in vivo* accounts for the peaks and strikes observed in Hi-C matrices, which are commonly translated as topological associated regions (TADs) in G1 cells (Fudenberg *et al*, 2016, Sanborn et al., 2015). Recent studies confirmed that cohesin-mediated loops and the positions of TADs emerge quickly after telophase by producing contact patterns consistent with a LE process (Abramo *et al*, 2019; Zhang *et al*, 2019).

We can postulate several non-excluding hypotheses to explain the different effects of condensin and cohesin on DNA knotting. One possibility could rely on the dynamics of their LE activity (Fig 7). Cohesin is likely to conduct discrete LE events to generate structural loops within specific boundaries. Subsequent stabilization of such loops would then favor intramolecular entanglement of DNA, as has been demonstrated with polymer simulations (Najafi & Potestio, 2015). Conversely, condensin may perform more dynamic rounds of LE without specific boundaries to scan the presence of DNA entanglements and promote their removal genome-wide. This scenario might be analogous to that occurs in mitotic chromatin, where cohesin may favor SCI formation by maintaining sister chromatids in close proximity (Sen *et al*, 2016; Goloborodko *et al*, 2016b; Piskadlo *et al*, 2017), whereas condensin might be performing continuous rounds of LE to enforce

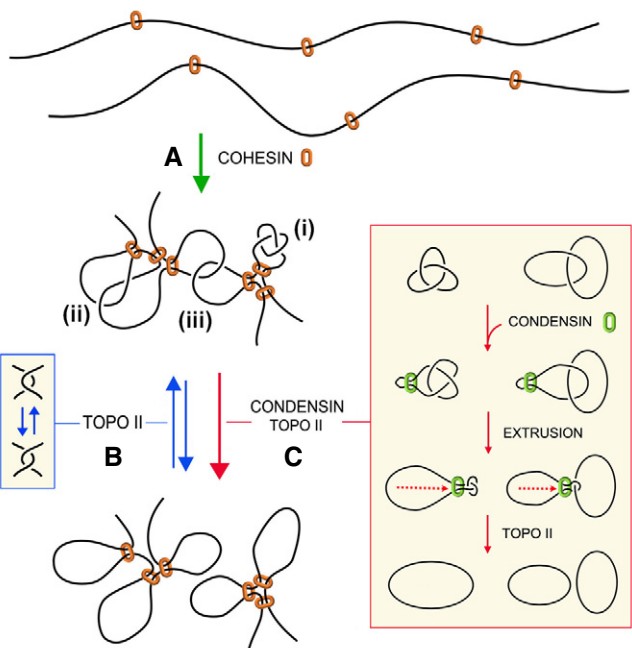

**Figure 7. Model of condensin role in the minimization of DNA entanglements.**

A  Cohesin generates and stabilizes DNA loops to organize interphase chromatin into topological domains.

B  Random DNA strand passage activity of topo II can either remove or produce DNA entanglements within and across such topological domains. Juxtapositions of DNA segments within a loop can lead to the formation of knots (i), whereas juxtapositions of DNA segments belonging to nearby loops or adjacent domains can lead to the formation of intra- (ii) or inter-molecular (iii) DNA interlinks.

C  To minimize the occurrence of these entanglements, condensin might use its DNA loop extrusion activity to constrict intra- and inter-molecular interlinks and so bias the DNA strand passage activity of topo II to remove them. This condensin function may operate during interphase to facilitate chromatin transactions and during cell division to enforce the removal of sister chromatid interlinks.

the removal of SCI. A second possibility could rely on distinct coordination of condensin and cohesin with topo II activity. Based on immunofluorescence and ChIP data, topo II occupies similar genomic loci to condensin and cohesin, but their functional interplay remains unknown. Some studies suggested that condensin can physically interact with topo II and stimulate its activity (Bhat *et al*, 1996; D'Ambrosio *et al*, 2008a). Yet, other studies have failed to confirm a physical interaction (Bhalla *et al*, 2002; Lavoie *et al*, 2002; Cuvier & Hirano, 2003) or a stimulatory effect (Charbin *et al*, 2014). Likewise, a physical or functional interaction of cohesin and topo II has been proposed, as both complexes colocalize at DNA loop boundaries (Uuskula-Reimand *et al*, 2016; Canela *et al*, 2017). Lastly, the distinct effects of condensin and cohesin could result from unequal binding to minichromosomes. This possibility, however, seems less likely considering the comparable abundance and broad chromosomal distribution of both complexes in budding yeast (Glynn *et al*, 2004; Wang *et al*, 2005). Accordingly, the effects of condensin and cohesin on $P^{kn}$ are accentuated with DNA length independently of the functional elements present

in the minichromosomes. Only the lack of effects observed upon the inactivation of the Smc5/6 complex could be attributed to the lower abundance of this complex in comparison with cohesin and condensin (Aragon, 2018).

### Role of condensin during interphase

The generally established essential function of condensin is the compaction and individualization of sister chromatids to facilitate their segregation during cell divisions (Strunnikov *et al*, 1995; Hirano *et al*, 1997). To this end, condensin might play both an active role in promoting the removal of SCI (Sen *et al*, 2016; Piskadlo *et al*, 2017) and a structural role in organizing the axial architecture of mitotic chromosomes (Maeshima & Laemmli, 2003; Ono *et al*, 2003; Walther *et al*, 2018). These mitotic roles are achieved by the single condensin complex found in yeast cells and by the two condensin complexes (condensin I and II) found in metazoans (Hirota *et al*, 2004; Hirano, 2012). However, former studies in budding yeast revealed that condensin is also present in interphase chromatin (Freeman *et al*, 2000; Lavoie *et al*, 2002), where it is distributed over the length of every chromosome throughout the cell cycle (Wang *et al*, 2005; D'Ambrosio *et al*, 2008b). Likewise, condensin II is also present in interphase chromatin in metazoans (Hirano, 2012; Frosi & Haering, 2015). The role of condensin during interphase is unknown, but its inactivation causes large-scale changes in the chromatin structure of budding yeast (Bhalla *et al*, 2002; Lazar-Stefanita *et al*, 2017; Paul *et al*, 2018). Inactivation of condensin II produces intermixing of chromosomal territories in Drosophila (Rosin *et al*, 2018; Rowley *et al*, 2019) and an increase of inter-chromosome associations in mammals (Nishide & Hirano, 2014). Other studies concur that condensin disruption alters a wide range of processes including gene regulation, DNA repair and recombination (Frosi & Haering, 2015; Paul *et al*, 2019). It is intriguing how so many functions and phenotypes are connected to condensin activity. According to our findings, the answer could be that condensin is promoting the removal of harmful DNA knots and interlinks that topo II activity might produce during topological equilibration of chromatin fibers and domains (Fig 7). Such DNA entanglements can alter, for instance, the progression of RNA polymerases and the assembly of nucleosomes, as demonstrated by *in vitro* studies (Portugal & Rodriguez-Campos, 1996; Rodriguez-Campos, 1996). Therefore, the failure of condensin to promote DNA untangling is expected to interfere with multiple genome transactions during interphase, in addition to the individualization of chromosomes during cell division.

The unanticipated role of condensin in minimizing DNA entanglements raises new questions, such as how the LE activities of condensin and cohesin may interplay with each other throughout the cell cycle. A similar issue arises in mitotic chromatin, in which the cohesion of sister chromatids, the removal of SCI, and DNA looping along the axial architecture of chromosomes involve the coordination of distinct SMC activities. Another relevant matter is the interplay of SMCs and type-2 topoisomerases, which are highly conserved from bacteria to eukaryotes. The coordination of these two essential machineries might have been primordial throughout evolution to minimize DNA entanglements as genomes increased in size and complexity.

## Materials and Methods

### DNA constructs and yeast strains

Plasmids YEp13, YEp24, YRp21, YCp50, YRp5, YRp4, and YRp3 (Appendix Fig S1) were amplified in *Escherichia coli* and, when indicated, converted into circular minichromosomes by transforming *Saccharomyces cerevisiae* using standard procedures (Valdes *et al*, 2018). Cellular topo II assays were done in the topo I-deficient strains JCW27 (*MATa, Δtop1, his3-D200, leu2-D1, trp1-D63, ura3–52*) and JCW28 (*MATa, Δtop1, top2-4, his3-D200, leu2-D1, trp1-D63, ura3–52*) (Trigueros & Roca, 2001). Condensin function was tested in AS330 (*MATa, smc2-8, ura3, leu2, lys2, his3, ade2*) (Freeman *et al*, 2000). The *smc2-8* mutation was introduced in yeast strains JCW25 (*MATa, his3-D200, leu2-D1, trp1-D63, ura3–52*) and its derivative JCW26 (*top2-4*) by two-step gene replacement involving the counter selectable marker *URA3* (Rothstein, 1991). Cohesin function was tested in the strain K5832 (*MATa, scc1-73, ade2-1, ura3–52, TRP+, can1-100, leu2-3, 112, his3-11*) (Michaelis *et al*, 1997). Smc5/6 function was tested in CCG1428 (*MATa, smc6-9, bar1Δ, leu2-3 112, ura3-52, his3-D200, trp1-D63, ade2-1, lys2-801, pep4*) (Torres-Rosell *et al*, 2005). Thermo-sensitivity of SMC complexes and topoisomerase mutants was corroborated by drop growth assays (Appendix Fig S2).

### Topo II activity in crude yeast lysates

To target cellular topo II activity with ICRF-193 (Sigma-Aldrich), JCW27 (*Δtop1*) cells bearing YEp13 were grown at 30°C in synthetic dropout -LEU media containing 2% glucose. Exponential 50 ml cultures (OD$_{600}$ = 0.6–0.8) were harvested and washed twice in TE (Tris–HCl 10 mM (pH 8) EDTA 1 mM) and resuspended at 4°C in 1 ml of lysis buffer (Tris–HCl 10 mM pH 8.0, EDTA 1 mM, EGTA 1 mM, NaCl 150 mM, DTT 1mM, Triton X-100 0.1%, pepstatin 1μg/ml, leupeptin 1μg/ml, PMSF 1 mM). Resuspended cells were transferred to 15-ml conic tubes and mixed with 1 ml of acid-washed glass beads (425–600 μm, Sigma). Mechanic lysis of> 80% cells was achieved by stirring six times with a vortex apparatus for 30 sec at 4°C. Glass beads and large cell debris were removed by centrifugation (2000 g x 2 min at 4°C). Cell lysates (0.5 ml) were supplemented with 5 mM MgCl$_2$ and 2 mM ATP and with 100 ng of a negatively supercoiled control plasmid (YEp24). Following incubation at 30°C for 20 min, ICRF-193 was added (100 μM) and incubation continued at 30°C for 10 min. Reactions were quenched by adding EDTA (20 mM) and SDS (0.2%) and extracted twice with phenol and once with chloroform. Nucleic acids were precipitated with ethanol and dissolved in 100 μl of TE containing RNAse-A. Following 10-min incubation at 37°C, ammonium acetate was added to 0.5 M and DNA was precipitated with ethanol. Each DNA sample was dissolved in 40 μl of TE prior gel electrophoresis. To test *Top2-Δ83* activity in yeast, JCW28 (*Δtop1 top2-4*) cells bearing YEp13 and the expression plasmids *pGAL1T2* or *pGAL1T2Δ83* (Martinez-Garcia *et al*, 2014) were grown at 26°C in synthetic dropout -URA -LEU media containing 2% glucose. Exponentially growing cultures were diluted to OD$_{600}$ = 0.1 in YEP containing 2% raffinose. When OD$_{600}$ = 0.6-0.8 was reached at 26°C, galactose was added to a 2% final concentration and the cell cultures were shifted to 35°C for 2 h. Cells were harvested and crude lysates were prepared by

stirring with glass beads as described above. A sample of the lysates was loaded in SDS–PAGE gels to confirm the extrachromosomal expression of *TOP2* and *Top2-Δ83* proteins. Upon addition of 5 mM $MgCl_2$, 2 mM ATP and 100 ng of negatively supercoiled plasmid YEp24, the lysates were incubated for 30 min at 35°C. Reactions were quenched and nucleic acids isolated for gel electrophoresis analyses as described above.

**SMC mutants culture and DNA extraction**

Yeast strains bearing distinct circular minichromosomes were grown at 26°C in the adequate synthetic dropout media supplemented with 2% glucose. Exponentially growing cultures $OD_{600}$ = 0.6–0.8 were maintained at 26°C or shifted to 35°C for 60 min to inactivate the temperature-sensitive alleles. To arrest the cells in G1, alpha-factor to a final concentration of 2 mg/L was added to exponentially growing cultures every 30 min for 2 h at 26°C and then for one additional hour upon shifting one half of the cultures to 35°C. To arrest the cells in metaphase, nocodazole was added to exponentially growing cultures to a final concentration of 15 mg/mL for 2 h at 26°C and then for one additional hour upon shifting one half of the cultures to 35°C. Following the inactivation of the temperature-sensitive alleles, the DNA topology of circular minichromosomes was fixed *in vivo* by quickly mixing the liquid cultures with one cold volume (−20°C) of ETol solution (Ethanol 95%, 28 mM Toluene, 20 mM Tris–HCl pH 8.8, 5 mM EDTA) (Diaz-Ingelmo *et al*, 2015). To measure cellular DNA content (1n, 2n), about $10^6$ of ETol fixed cells were washed with saline-sodium citrate (SSC) buffer, incubated for 1 h at 37°C in SSC containing 0.1 mg/mL RNase-A and again incubated for 1 h at 50°C in SSC containing 1 mg/mL Proteinase K. Cell samples in 1 mL SSC were sonicated for two 30 sec cycles at 4C and incubated at 25°C for 1h in presence of 3 mg/mL propidium iodide prior flow cytometry reading on a Gallios (Beckman Coulter) cell analyzer. To extract total DNA, ETol fixed cells from 25 ml cultures were sedimented, washed twice with TE, resuspended in 400μl of TE, and transferred to a 1.5-ml microfuge tube containing 400μl of phenol and 400μl of acid-washed glass beads (425–600 μm, Sigma). Mechanic lysis of> 80% cells was achieved by shaking the tubes in a FastPrep® apparatus for 10 sec at power 5. The aqueous phase of the cell lysates was collected, extracted with chloroform, precipitated with ethanol, and dissolved in 100 μl of TE containing RNAse-A. Following 10-min incubation at 37°C, ammonium acetate was added to 0.5 M and DNA was precipitated with ethanol. Each DNA sample was dissolved in 40 μl of TE prior gel electrophoresis.

**DNA electrophoresis for topology analyses**

Lk distributions of control plasmid YEp24 were examined with 1D-electrophoreses carried out in 0.8% agarose gels in TBE buffer (89 mM Tris-borate, 2 mM EDTA) plus 0.2 μg/ml of chloroquine and run at 50V for 14 h. Lk distribution of minichromosomes YEp13, YRp21, and YCp50 were examined with 2D-electrophoreses carried out in 0.6% agarose gels (20 × 20 cm) in TBE buffer plus 0.6 μg/ml of chloroquine in the first dimension (30V for 36 h) and in TBE buffer plus 3 μg/ml of chloroquine in the second dimension (80V for 4 h). 2D electrophoreses of YRp3, YRp4, YRp5, and 2-micron circles were carried out in 0.8% agarose gels (20 × 20 cm)

in TBE buffer plus 0.6 μg/ml of chloroquine in the first dimension (50V for 14 h) and TBE buffer plus 3 μg/ml of chloroquine in the second dimension (60V for 6 h).

To examine the DNA knots formed in the minichromosomes, DNA samples were nicked with endonuclease BstNBI (NEB). 2D-electrophoreses of nicked DNA of YRp3, YRp4, and YRp5 circles were carried out in a 0.9% agarose gel (20x20 cm) in TBE buffer at 33V for 40 h in the first dimension and at 150V for 3 h in the second dimension. 2D-electrophoreses of nicked DNA of 2-micron and YCp50 circles were carried out in a 0.6% agarose in TBE buffer at 25V for 40 h in the first dimension and at 125V for 4 h in the second dimension. 2D-electrophoreses of nicked DNA of YEp13 and YRp21 circles were carried out in a 0.4% agarose in TBE buffer at 25V for 40 h in the first dimension and at 125V for 4 h in the second dimension.

All 2D-gels were blot-transferred to positively charged nylon membranes (Hybond-N$^+$, Amersham Biosciences). Blots were hybridized with minichromosome DNA probes labeled with AlkPhos Direct (GE Healthcare®). Probe signals were visualized following incubation with CDP-Star detection reagent (GE Healthcare®) for 10 min at room temperature and recorded on X-ray films. DNA knot probability ($P^{Kn}$) was calculated as described previously (Valdes *et al*, 2018) by quantifying non-saturated signals obtained with serial dilutions of DNA knot samples or with different exposure periods, using the ImageJ software. $P^{Kn}$ values are the relative abundance of total knot populations with respect to the total amount of unknotted and knotted DNA circles.

**Expanded View** for this article is available online.

## Acknowledgements

We would like to thank the laboratories of Kim Nasmyth, Jordi Torres, and Luis Aragón for sharing yeast strains. We would also like to thank Martí Aldea, David Moreno, and Jonathan Baxter for their insightful input and discussion. This research was supported by the Plan Estatal de Investigación Científica y Técnica of Spain, with grants BFU2015-67007-P and PID2019-109482GB-I00 to J.R; and research fellowships BES-2016-077806 to S.D., BES-2012-061167 to J.S., and BES-2015-071597 to A.V.

## Author contributions

Experiments: JR, SD, JS, BM-G, AV; Data analysis: JR, SD; Manuscript writing: JR, SD, AV.

## Conflict of interest

The authors declare that they have no conflict of interest.

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
