## [Review Process File · The EMBO Journal]

Condensin minimizes topoisomerase II-mediated entanglements of DNA in vivo

Sílvia Dyson, Joana Segura, Belén Martínez-García, Antonio Valdés, and Joaquim Roca

DOI: [10.15252/embj.2020105393](https://doi.org/10.15252/embj.2020105393)

Corresponding author(s): Joaquim Roca (jrbmc@ibmb.csic.es)

Review Timeline:

Submission Date:	23rd Apr 20
Editorial Presonsultation:	22nd Jun 20
Author Response:	29th Jun 20
Editorial Decision:	2nd Jul 20
Revision Received:	10th Sep 20
Accepted:	7th Oct 20

Editor: Hartmut Vodermaier

Transaction Report:

Thank you for submitting your manuscript on condensin/cohesin roles in DNA entanglements in yeast to The EMBO Journal. I am very sorry for the considerable delay in getting back to you with a response - your manuscript had been sent to three expert reviewers, of whom one -despite multiple reminders from our side- has still not returned their comments. With the reports of the other two reviewers not being black-and-white, I had been hoping for the outstanding report to come in, but decided to now forward the two available reports to you already, in order to prevent further loss of time.

As you will see, both reviewers generally appreciate the potential interest of your results, particularly in light of the overall interest of the topic, but they also bring up a number of major concerns, including important issues with experimental descriptions and data presentation, that would have to be satisfactorily addressed before publication may be warranted. Since it is currently difficult to assess whether these uncertainties can be easily clarified, which would allow us to make more definitive commitments regarding eventual acceptance at The EMBO Journal, I would appreciate hearing from you how you would envision addressing the various points of the referees in the case of a revision. Therefore, please carefully consider the attached reports and send back a brief point-by-point response outlining how the referees' comments might be addressed/clarified. These tentative response would be taken into account when making our final decision on this manuscript, and could also serve as a basis for further direct discussions. It would be great if you could get back to me with such a response ideally by the beginning of next week.

Referee #1:

Top2 is an essential enzyme for genome stability due to its unique ability to resolve sister chromatid intertwines and relax DNA molecules. However, it has long been acknowledged that under high DNA concentrations this enzyme is able to catalyze the reverse reaction (i.e promote DNA intertwines/knots). How this dangerous activity is prevented in vivo has remained mysterious, mostly due to the difficulty in assessing DNA topology of chromatinized DNA inside living cells. In this very interesting manuscript by Dyson and co-workers, the authors provide convincing and important findings to elucidate how the topology state of chromatin is modulated by topoisomerase

2 in vivo. The authors describe that reduced knotting probability does not rely on an intrinsic capacity of top2 to simplify topology (as has been widely argued based on in vitro evidence) but is instead regulated by SMC complexes. In particular, condensin complexes are proposed to reduce the knotting probability whereas cohesins display the opposite effect. The reported findings are novel and of major importance for chromosome biology.

The authors take advantage of a recently developed experimental approach that allows the visualization of DNA topology states of chromatinized circular minichromosomes (Valdes et al 2018), thereby surpassing current limitations on measuring DNA topology in vivo. In their prior study, it was demonstrated that chromatin can form intramolecular knots with the intriguing observation that these do not scale with chromosome size, suggesting additional mechanisms reduce knot occurrence. In this manuscript the authors now elucidate (at least some) mechanisms that regulate the probability of intramolecular knots.

SMC complexes are emerging as major players in chromosome architecture with wide implications in all aspects involving the DNA molecule (transcription, replication, repair, segregation). Therefore, the findings presented here are of very wide interest. They bring important concepts of topoisomerase enzymology, mostly studied in vitro, to the context of real chromatin, thereby filling a major gap in the topology field. Although prior studies have previously suggested how SMC complexes can modulate the end result of topoisomerase 2 reactions (reviewed in Piskadlo and Oliveira *nt. J. Mol. Sci.* 2017.), these were mostly based on indirect observations of DNA topology (mostly the presence of sister chromatid intertwinings in mitosis). The present study provides a much more detailed analysis on how these complexes functionally interact in the architecture of chromatin throughout the cell cycle, based on direct topology measurements, thereby demonstrating how these interactions can impact on overall genome architecture even outside mitosis. I am therefore in favour of its publication.

There are, however, a few points that I would like to see addressed before acceptance.

Major points:

1. Although the manuscript is very well written, the authors should make an effort to make it more accessible to readers that are not so familiar with topology analysis. Considering the wide implications of these important findings, and the broad readership of EMBO Journal, some results are not of immediate understanding for a non-topology expert. Maybe some additional drawings next to the plots or further explanations in the text may help (for example, why a narrow profile on Lk distribution reflects changes in the simplification ability? how ICRF-193 treatment impairs topology simplification without impairing its relaxation capacity? How is Pkn calculated (a simplified description in the text would help instead of having to read M&M)?)
2. The results are indeed consistent with a loop extrusion activity by condensin. However, this was not directly addressed in the present study. The authors should therefore tone down their interpretations (e.g. page 4 "these results provide evidence that the DNA loop extrusion of condensin is functional in vivo and that it works to minimize the entanglement of intracellular DNA"). The results clearly demonstrate that condensin reduces knotting probability but whether this is a direct consequence of LE remains to be addressed. Note that cohesin also extrudes loops so something else must ensure this condensin-dependent activity. In line with this, the drawing presented on Fig. 1F would probably be better placed at the end, as a proposed model, referring that this would refer to condensin action and present a similar model for how cohesin could have the opposite role. As it stands, it is too hypothetical to be a starting point of the manuscript.
3. The opposing role of cohesin is less obvious to understand (and indeed the authors did not even include it in the final model (Fig. 7). One possibility would be that sister chromatid cohesion alone

decreases intramolecular knotting, simply by reducing intramolecular contacts. However, the same reduction was detected in G1 cells so the effect must be independent of sister chromatid cohesion (which should be discussed). The authors discuss the possibility that this could be due to changes in loop extrusion dynamics but this was never tested. What would happen if the same experiments were performed in the absence of *wapl/rad61*, where cohesin should be more stably associated and possibly able to extrude larger loops? As it stands, the manuscript title/abstract should probably reflect an emphasis on the most convincing data (regarding condensin) and tone down the conclusions regarding cohesin.

4. Throughout the manuscript the authors present several graphs with quantification of Pkn as mean \pm SD, although no statistical analysis was ever performed. It would be important to include some statistics as the effect of cohesin is indeed rather small and unclear whether or not it is significant.

Minor points:

1. The notion that SMC inactivation could lead to disruption in the equilibrium of top2 reactions has been proposed before, in the context of sister chromatid resolution (Sen et al 2016; Piskadlo et al 2017). This should be mentioned earlier in the introduction rather than simply in the discussion.
2. Figures could be improved to have a more immediate read-out of the experimental layout. For example, in Fig. 2B, it would be good to have directly in the figure what 1, 2, 3 and 4 are; also, it would be good to have a graphical correspondence between the gels and the intensity profiles, either by colouring the respective numbers or by assigning the corresponding numbers to the profiles. The same applies to Fig. 2E, (clarify what 1 and 2 are and provide some sort of graphical correspondence between the gels and the intensity profiles).
3. The identity of the control plasmid used should be mentioned (it is solely referred as "8 Kb negatively supercoiled DNA plasmid"), even if only in figure legend or M&M.
4. On Page 6, is the top2-d83 form expressed in levels identical to those of top2? Or, in other words, is the concentration/amount of each in the crude lysate similar?
5. On page 7, correct Pkn=0.22 (I believe it must be 0.022)
6. For the cell cycle specific analysis, it would be advantageous to clarify that temperature shift was performed after the arrest somewhere in the figure, figure legend or results description. I had to read the M&M section to really understand the experimental layout.
7. Regarding the size dependence, the "small" minichromosome used in Fig. 4, 5 and 6 are not always consistent (e.g. YRp5 is only used in Fig. 5 and YRp4 is not used in this figure). Is there any reason for that?
8. Page 11 - *drosophila* should be corrected to *Drosophila*.
9. The paragraph on the "Role of condensin during interphase" should be rephrased to clarify particular differences between yeast and metazoans. For example, the authors state that condensin is bound on equal amounts in interphase vs mitosis in yeast. But in metazoans only condensin II is bound in interphase hence mitotic chromosomes have far larger amounts of condensins. Accordingly, the chromosome territories studies pointed out refer to condensin II.
10. The sentence "In fact, this scenario already occurs in sister chromatids during metaphase, in which condensin promotes SCI removal and so counteracts the effect of cohesin that favors SCI formation by maintaining sister chromatids in close proximity" (page 10) should be followed by the appropriate references.
11. The subsequent sentence states that condensin interacts with top2. A direct physical interaction has been proposed in Bhat et al, but several studies have failed to confirm this observation (see for example, Lavoie et al 2000; Bhalla et al 2002; Cuvier & Hirano 2003). There is indeed a lot of evidence that they occupy similar genomic loci (based on immunofluorescence and ChIP data) but the most accepted view, to my knowledge, is that this does not rely on a direct

- physical interaction. Additionally, whether condensin stimulates top2 activity has also remained controversial (see Charbin et al NAR 2014). This sentence should be rephrased accordingly.
12. The findings that condensin depletion do not lead to changes in DNA supercoiling are indeed very interesting as they exclude a prior model to explain the functional interplay between condensin and top2 (proposed in Baxter et al). It would be interesting to refer to recent studies that also describe that condensin compacts DNA at the same rate, independently of the topological state of the DNA (Eeftens et al EMBO J 2017).
 13. throughout the manuscript, but especially in the methods section, when abbreviating for degrees Celsius authors alternate between C, °C oC. One single form should be adopted and used in the entire manuscript.
 14. the description of yeast cell lysis in "Culture of yeast SMC mutants and DNA extraction" is good, as it allows for reproducibility. The same degree of detail should be used in "Topo II activity in crude yeast lysates"
 15. raffinose should be corrected to raffinose
 16. Why were top ts mutants incubated at 30°C for the restrictive temperature instead of 35°C?
 17. correct 50-ml to 50 ml
 18. page 11 replace "transfected" by "transformed" (the most commonly used expression for yeast)
 19. confirm K5832 genotype.

Referee #3:

Reviewer report:

This study investigated the generation and resolution of DNA knots by topoisomerase II, condensin, cohesin and the smc5/6 complex using plasmids and minichromosomes of different sizes that were transfected into yeast cells.

The main conclusion is that inactivation of condensin increases the knots in a minichromosomes, therefore concluding that condensin's loop extrusion activity might have a role in resolving DNA knots.

This observation is very interesting and would contribute to our understanding how cohesin and condensin organize chromatin during mitotic chromosome compaction and the role of condensin on interphase chromatin.

However, there are a number of experimental and presentation issues that need to be addressed. In its current state, the manuscript, specifically the results section, is written for a very specific readership in the field and not suitable for the broad readership of the EMBO Journal.

The authors should use graphics in the figures, eg. Figure S2, to explain the experimental results. For readers that are not experienced in this type of analyses it will be very difficult to appreciate the information, in particular on the linkage analysis figures. There is a great variation between these graphs in different figures, which is probably the nature of the experiment. However, it would be very helpful to see a "positive control" that shows which type of change can be expected in the figure when linkage is changed. Also, very specific terminology, eg. "thermal Lk distribution" or "precluding the oversimplification capacity of topo II" should be introduced or simplified. Also, the authors discuss several times "negative Lk values" but there is no quantitation of this somewhere in the manuscript. Further, the authors should thoroughly revise their manuscript and avoid overly complicated language as in this sentence: "Our results show that disrupting the simplification activity of cellular topo II does not upturn the low DNA knotting probability of chromatin.". Why not just say "... increase the DNA knotting ability ..". There are also a number of grammar problems, eventually proofreading by a native speaker could help.

Major concerns:

Any documentation that the used yeast strains have the expected protein deletion under the experimental conditions, for example a western blot with suitable antibodies or functional assays, is missing.

In Figure 2B the authors want to demonstrate that the incubation with topo II has an effect on the Lk distribution and that this effect disappears upon addition of a topoisomerase inhibitor.

Unfortunately, the effect cannot be appreciated in this figure. The distribution in lanes 2-4 looks nearly the same. If there is a significant change the authors should highlight this. The intensity plots should be better labelled and brought in context with the gel lanes. If the inhibitor treatment makes lane 4 more similar to lane 2, also these intensity plots should be compared.

In Figure 2 C the authors resolve the linking distribution in a 2D gel to establish the difference with inhibitor treatment. They state "Following the addition of ICRF-193, the Lk distribution of YEp13 was not significantly altered, in sharp contrast to that observed in the control plasmid." It is unclear to which observation or quantification in the blot the authors refer to. A visual impression of the graph seems to be insufficient in the eyes of this reviewer.

In the text the authors state: "Before adding ICRF-193, YEp13 presented a distribution of topoisomers of negative ΔLk values (Fig S2)", I cannot see this in this figure.

Then the authors nick the DNA to analyze knots and state "Following the addition of ICRF-193, the knot probability of YEp13 was not significantly altered ($P_{kn} \approx 0.04$).". To state this the authors should perform a statistics test (also for all other experiments containing quantifications).

Also in Figure 2F in the upper panel it is not clear which change is anticipated if the effect would be positive. As suggested already above, can the authors include a positive control?

In Figures 4, 5 and 6 the authors investigate the effect of cohesin, condensin and smc5/6 inactivation on plasmids and minichromosomes of different size, containing also different functional elements. Is there evidence that cohesin, condensin and smc5/6 associate with these plasmids and could the number of proteins associated/ binding sites present account for the outcome of the experiments?

The model in Figure 7 illustrates, how condensin could promote the removal of DNA knots. However, also the "loop extrusion" part of the model requires topo II. So, if the model is correct, one would have expected an effect of the topo II inactivation on the knotting efficiency in Fig. 2. An experiment using the double mutant should demonstrate whether this activity of condensin is indeed topo II dependent.

Minor suggestion:

In the introduction Fig.1, the authors mention also catenanes that might be present in G2 phase cells. Can these be observed in those assays?

Response to referee reports for EMBOJ-2020-105393

Referee #1

Major points:

1. Although the manuscript is very well written, the authors should make an effort to make it more accessible to readers that are not so familiar with topology analysis. Considering the wide implications of these important findings, and the broad readership of EMBO Journal, some results are not of immediate understanding for a non-topology expert. Maybe some additional drawings next to the plots or further explanations in the text may help (for example, why a narrow profile on Lk distribution reflects changes in the simplification ability? how ICRF-193 treatment impairs topology simplification without impairing its relaxation capacity? How is Pkn calculated (a simplified description in the text would help instead of having to read M&M)?)

We agree. We will include additional drawings/schemes in the figures to allow quick understanding of the experiments for non-topology experts. We will include further explanations in the results section and/or figure legends, as well as additional supplementary figures to clarify the meaning of narrowed Lk distributions, the mechanism of ICRF-193, and how is Pkn calculated.

2. The results are indeed consistent with a loop extrusion activity by condensin. However, this was not directly addressed in the present study. The authors should therefore tone down their interpretations (e.g. page 4 "these results provide evidence that the DNA loop extrusion of condensin is functional in vivo and that it works to minimize the entanglement of intracellular DNA"). The results clearly demonstrate that condensin reduces knotting probability but whether this is a direct consequence of LE remains to be addressed. Note that cohesin also extrudes loops so something else must ensure this condensin-dependent activity. In line with this, the drawing presented on Fig. 1F would probably be better placed at the end, as a proposed model, referring that this would refer to condensin action and present a similar model for how cohesin could have the opposite role. As it stands, it is too hypothetical to be a starting point of the manuscript.

We agree. We will tone down the interpretation of the results, indicating that they are just consistent with DNA loop extrusion of condensin in vivo. As suggested, we will modify the drawing on Fig. 1F, and develop the model on Fig. 7 to illustrate opposite effects of condensin and cohesin.

3. The opposing role of cohesin is less obvious to understand (and indeed the authors did not even include it in the final model (Fig. 7)). One possibility would be that sister chromatid cohesion alone decreases intramolecular knotting, simply by reducing intramolecular contacts. However, the same reduction was detected in G1 cells so the effect must be independent of sister chromatid cohesion (which should be discussed). The authors discuss the possibility that this could be due to changes in loop extrusion dynamics but this was never tested. What would happen if the same experiments were performed in the absence of wapl/rad61, where cohesin should be more stably associated and possibly able to extrude larger loops? As it stands, the manuscript title/abstract should probably reflect an emphasis on the most convincing data (regarding condensin) and **tone down** the conclusions regarding cohesin.

Certainly, the effects of cohesin are not strong and are more difficult to interpret. As suggested, we will tone down the conclusions regarding cohesin and put the emphasis on the most convincing data of condensin. In this respect, we already considered modifying the title since the wording "Condensin counteracts cohesin" is misleading. More appropriated titles could be:

Condensin drives topoisomerase II to minimize in vivo DNA entanglements

Condensin minimizes topoisomerase II-mediated entanglement of intracellular DNA

4. Throughout the manuscript the authors present several graphs with quantification of Pkn as mean \pm SD, although no statistical analysis was ever performed. It would be important to include some statistics as the effect of cohesin is indeed rather small and unclear whether or not it is significant.

We agree. We will show the P values in the plots, which will confirm that the effect of cohesin is rather small.

Minor points:

1. The notion that SMC inactivation could lead to disruption in the equilibrium of top2 reactions has been proposed before, in the context of sister chromatid resolution (Sen et al 2016; Piskadlo et al 2017). This should be mentioned earlier in the introduction rather than simply in the discussion.

We agree. We will move these two references earlier in the introduction

2. Figures could be improved to have a more immediate read-out of the experimental layout. For example, in Fig. 2B, it would be good to have directly in the figure what 1, 2, 3 and 4 are; also, it would be good to have a graphical correspondence between the gels and the intensity profiles, either by colouring the respective numbers or by assigning the corresponding numbers to the profiles. The same applies to Fig. 2E, (clarify what 1 and 2 are and provide some sort of graphical correspondence between the gels and the intensity profiles).

We agree. Fig 2B and 2E are hard to follow. As suggested, we will include experimental layouts and graphical correspondences.

3. The identity of the control plasmid used should be mentioned (it is solely referred as "8 Kb negatively supercoiled DNA plasmid"), even if only in figure legend or M&M.

We will identify the control plasmid.

4. On Page 6, is the top2-d83 form expressed in levels identical to those of top2? Or, in other words, is the concentration/amount of each in the crude lysate similar?

Yes. We will remark that both proteins are expressed in comparable amount.

5. On page 7, correct Pkn=0.22 (I believe it must be 0.022).

Thanks. We will correct this typo.

6. For the cell cycle specific analysis, it would be advantageous to clarify that temperature shift was performed after the arrest somewhere in the figure, figure legend or results description. I had to read the M&M section to really understand the experimental layout.

We will indicate in the figures and results how temperature shifts were performed.

7. Regarding the size dependence, the "small" minichromosome used in Fig, 4, 5 and 6 are not always consistent (e.g. YRp5 is only used in Fig. 5 and YRp4 is not used in this figure). Is there any reason for that?

Yes. We will indicate that YRp5 was used instead of YRp4 because the strain was TRP+ and so we could not use this auxotrophic marker.

8. Page 11 - drosophila should be corrected to Drosophila.

Ok

9. The paragraph on the "Role of condensin during interphase" should be rephrased to clarify particular differences between yeast and metazoans. For example, the authors state that condensin is bound on equal amounts in interphase vs mitosis in yeast. But in metazoans only condensin II is bound in interphase hence mitotic chromosomes have far larger amounts of condensins. Accordingly, the chromosome territories studies pointed out refer to condensin II.

We agree. The paragraph will be rephrased as suggested.

10. The sentence "In fact, this scenario already occurs in sister chromatids during metaphase, in which condensin promotes SCI removal and so counteracts the effect of cohesin that favors SCI formation by maintaining sister chromatids in close proximity" (page 10) should be followed by the appropriate references.

We will add the missing references.

11. The subsequent sentence states that condensin interacts with top2. A direct physical interaction has been proposed in Bhat et al, but several studies have failed to confirm this observation (see for example, Lavoie et al 2000; Bhalla et al 2002; Cuvier & Hirano 2003). There is indeed a lot of evidence that they occupy similar genomic loci (based on immunofluorescence and ChIP data) but the most accepted view, to my knowledge, is that this does not rely on a direct physical interaction. Additionally, whether condensin stimulates top2 activity has also remained controversial (see Charbin et al NAR 2014). This sentence should be rephrased accordingly.

We agree. We will rephrase this sentence and comment the suggested references.

12. The findings that condensin depletion do not lead to changes in DNA supercoiling are indeed very interesting as they exclude a prior model to explain the functional interplay between condensin and top2 (proposed in Baxter et al). It would be interesting to refer to recent studies that also describe that condensin compacts DNA at the same rate, independently of the topological state of the DNA (Eeftens et al EMBO J 2017).

We agree. We will add and comment this reference.

13. throughout the manuscript, but especially in the methods section, when abbreviating for degrees Celsius authors alternate between C, °C oC. One single form should be adopted and used in the entire manuscript.

Ok.

14. the description of yeast cell lysis in "Culture of yeast SMC mutants and DNA extraction" is good, as it allows for reproducibility. The same degree of detail should be used in "Topo II activity in crude yeast lysates"

Ok. We will explain this procedure in more detail in M&M.

15. raffinose should be corrected to raffinose

Ok.

16. Why were top ts mutants incubated at 30°C for the restrictive temperature instead of 35°C?

Thanks. We will correct this to 35°C.

17. correct 50-ml to 50 ml

Ok.

18. page 11 replace "transfected" by "transformed" (the most commonly used expression for yeast)

Ok.

19. confirm K5832 genotype.

Ok.

Referee #3

The authors should use graphics in the figures, eg. Figure S2, to explain the experimental results. For readers that are not experienced in this type of analyses it will be very difficult to appreciate the information, in particular on the linkage analysis figures. There is a great variation between these graphs in different figures, which is probably the nature of the experiment. However, it would be very helpful to see a "positive control" that shows which type of change can be expected in the figure when linkage is changed. Also, very specific terminology, eg. "thermal Lk distribution" or "precluding the oversimplification capacity of topo II" should be introduced or simplified. Also, the authors discuss several times "negative Lk values" but there is no quantification of this somewhere in the manuscript. Further, the authors should thoroughly revise their manuscript and avoid overly complicated language as in this sentence: "*Our results show that disrupting the simplification activity of cellular topo II does not upturn the low DNA knotting probability of chromatin.*". Why not just say "... increase the DNA knotting ability ..". There are also a number of grammar problems, eventually proofreading by a native speaker could help.

We agree. In the current state, some terminology and figures might be hard to follow by readers that are not experienced in DNA topology analyses. Therefore, we will include additional drawings/schemes in the figures to allow quick understanding of the significance of the results and controls. We will clarify in the text and/or in supplementary figures the meaning of "oversimplification capacity", "thermal Lk distribution", "negative Lk values". We will revise the manuscript to avoid complicated sentences.

Any documentation that the used yeast strains have the expected protein deletion under the experimental conditions, for example a western blot with suitable antibodies or functional assays, is missing.

We did not include these analyses because the used yeast strains have been well characterized in other studies. However, as requested, we can show in supplementary data functional assays (for example, growth phenotypes) that corroborate these mutant strains behave as expected.

In Figure 2B the authors want to demonstrate that the incubation with topo II has an effect on the Lk distribution and that this effect disappears upon addition of a topoisomerase inhibitor. Unfortunately, the effect cannot be appreciated in this figure. The distribution in lanes 2-4 looks nearly the same. If there is a significant change the authors should highlight this. The intensity plots should be better labelled and brought in context with the gel lanes. If the inhibitor treatment makes lane 4 more similar to lane 2, also these intensity plots should be compared.

We agree. As also indicated by review #1, presentation of results in figure 2 needs to be improved. Certainly, the Lk distributions in lanes 2-4 in figure 2B are virtually the same, in contrast to that in lane 3. We will include intensity plots of these three lanes and proper labels to clarify the effect of the inhibitor.

In Figure 2 C the authors resolve the linking distribution in a 2D gel to establish the difference with inhibitor treatment. They state "Following the addition of ICRF-193, the Lk distribution of YEp13 was not significantly altered, in sharp contrast to that observed in the control plasmid." It is unclear to which observation or quantification in the blot the authors refer to. A visual impression of the graph seems to be insufficient in the eyes of this reviewer.

We agree. Our statement could be confusing. We will include intensity plots of the Lk distributions of YEp13 minichromosome to allow direct comparison with the corresponding plots of the control plasmid.

In the text the authors state: "Before adding ICRF-193, YEp13 presented a distribution of topoisomers of negative ΔLk values (Fig S2)", I cannot see this in this figure.

We agree. We will describe the position of negative ΔLk values of YEp13 in Figure 2 and in Fig S2.

Then the authors nick the DNA to analyze knots and state "Following the addition of ICRF-193, the knot probability of YEp13 was not significantly altered ($P_{kn}=0.04$)." To state this the authors should perform a statistics test (also for all other experiments containing quantifications).

As also indicated by reviewer #1, we will show the statistic tests in the plots.

Also in Figure 2F in the upper panel it is not clear which change is anticipated if the effect would be positive. As suggested already above, can the authors include a positive control?

The positive control for this experiment is indeed the naked plasmid present in the reaction. We will include intensity plots in Figure 2F and/or supplementary figures to illustrate that there are no changes in the Lk distribution of the minichromosome, whereas the control plasmid undergoes broadening of the LK distribution.

In Figures 4, 5 and 6 the authors investigate the effect of cohesin, condensin and smc5/6 inactivation on plasmids and minichromosomes of different size, containing also different functional elements. Is there evidence that cohesin, condensin and smc5/6 associate with these plasmids and could the number of proteins associated/ binding sites present account for the outcome of the experiments?

Yeast minichromosomes have been used in many functional studies of the SMC complexes. Taking into account the abundance and broad genomic distribution of condensin and cohesin, there is no evidence to suspect that our results could be attributed to anything else. We only see the changes in knotting probability once we inactivate these complexes and these distinctive changes are reproduced irrespectively of the size, copy number and functional elements of the minichromosomes. Only in the case of smc5/6, which is less abundant than condensin and cohesin, we observe no significant changes. So, we will comment in the discussion that the no effect of smc5/6 may be also reflecting its lower abundance.

The model in Figure 7 illustrates, how condensin could promote the removal of DNA knots. However, also the "loop extrusion" part of the model requires topo II. So, if the model is correct, one would have expected an effect of the topo II inactivation on the knotting efficiency in Fig. 2. An experiment using the double mutant should demonstrate whether this activity of condensin is indeed topo II dependent.

We agree. Although topo II is likely the only activity able to knot/un knot DNA, the experiment proposed by the reviewer should unambiguously demonstrate whether knot minimization of condensin is indeed topo II dependent. We will conduct and include the experiment with the double mutant in our revised manuscript.

Minor suggestion:

In the introduction Fig.1, the authors mention also catenanes that might be present in G2 phase cells. Can these be observed in those assays?

By changing the electrophoretic conditions, we could observe bands compatible with catenanes in the upper part of the gels. These post-replication catenanes have been well characterized in previous studies (Baxter et al., 2011; Sen et al., 2016). We focused thus in the presence of DNA knots, which more directly reflect the spontaneous intramolecular entanglement of chromatin.

Thank you for your response letter and proposal for addressing the points raised by our two referees. which I have now had a chance to consider. I am happy to say that I found your explanations and clarifications well-taken, and likely to satisfy the referees' key concerns. I am therefore now formally inviting you to revise the study along these lines, and to resubmit a new version using the link below. Please note that it is our policy to allow only a single round of (major) revision, making it important to carefully revise and answer all points raised to the referees' satisfaction at this point.

Responses to reviewers' comments**Referee #1****Major points:**

1. Although the manuscript is very well written, the authors should make an effort to make it more accessible to readers that are not so familiar with topology analysis. Considering the wide implications of these important findings, and the broad readership of EMBO Journal, some results are not of immediate understanding for a non-topology expert. Maybe some additional drawings next to the plots or further explanations in the text may help (for example, why a narrow profile on Lk distribution reflects changes in the simplification ability? how ICRF-193 treatment impairs topology simplification without impairing its relaxation capacity? How is Pkn calculated (a simplified description in the text would help instead of having to read M&M)?)

We agree. To facilitate immediate understanding for non-topology experts, we did the following:

In the introduction, we describe in more detail the topo II mechanism (paragraph 3 and Fig 1C-D) and provided two new figures (Fig EV1 and EV2) to explain the ability to simplify the equilibrium DNA topology of topo II. Figure EV1 clarifies why a narrow profile on Lk distribution reflects changes in the simplification activity. Figure EV2 describes how ICRF-193 and the T2Δ83 enzyme might impair DNA topology simplification.

In the first section of the results -Topoisomerase II does not minimize the knotting probability of chromatin-, we improved the explanation and presentation of the experiments in figure 2 and indicate how Pkn is calculated. We replaced figure S2 with two new figures (Fig EV3 and EV4), which illustrate the experimental layout and interpretation of 2D gel electrophoresis of the DNA linking number topoisomers (Fig EV3) and the 2D gel electrophoresis of DNA knots (Fig EV4).

2. The results are indeed consistent with a loop extrusion activity by condensin. However, this was not directly addressed in the present study. The authors should therefore tone down their interpretations (e.g. page 4 "these results provide evidence that the DNA loop extrusion of condensin is functional *in vivo* and that it works to minimize the entanglement of intracellular DNA"). The results clearly demonstrate that condensin reduces knotting probability but whether this is a direct consequence of LE remains to be addressed. Note that cohesin also extrudes loops so something else must ensure this condensin-dependent activity. In line with this, the drawing presented on Fig. 1F would probably be better placed at the end, as a proposed model, referring that this would refer to condensin action and present a similar model for how cohesin could have the opposite role. As it stands, it is too hypothetical to be a starting point of the manuscript.

We agree. We cannot conclude that condensin reduction of knot probability is a direct consequence of LE. Therefore, we have toned down this notion in the abstract, the introduction (last paragraph) and the discussion (third paragraph) by specifying that our experimental observations are only consistent or support the LE activity of condensin *in vivo*.

Also, as suggested, we have placed the drawing in which DNA tangles are minimized by means of LE (initially presented on Figure 1F) to the final model in the discussion (Figure 7). In this model, we have also illustrated the plausible implication of cohesin to favor DNA entanglement.

3. The opposing role of cohesin is less obvious to understand (and indeed the authors did not even include it in the final model (Fig. 7). One possibility would be that sister chromatid cohesion alone decreases intramolecular knotting, simply by reducing intramolecular contacts. However, the same reduction was detected in G1 cells so the effect must be independent of sister chromatid cohesion (which should be discussed). The authors discuss the possibility that this could be due to changes in loop extrusion dynamics but this was never tested. What would happen if the same experiments were performed in the absence of wapl/rad61, where cohesin should be more stably associated and possibly able to extrude larger loops? As it stands, the manuscript title/abstract should probably reflect an emphasis on the most convincing data (regarding condensin) and tone down the conclusions regarding cohesin.

We agree. The effect of cohesin is small and difficult to interpret. Therefore, we revised the abstract and discussion section to put the emphasis on the most convincing data regarding condensin. In the discussion we comment that the plausible implication of cohesin on knot formation must be independent of its role in sister chromatid cohesion. We only postulate then why the effects of cohesin and condensin on P^{kn} are dissimilar. As also suggested by the editor, we changed the title of the manuscript to: "*Condensin minimizes topoisomerase II-mediated entanglements of DNA in vivo*".

4. Throughout the manuscript the authors present several graphs with quantification of P^{kn} as mean \pm SD, although no statistical analysis was ever performed. It would be important to include some statistics as the effect of cohesin is indeed rather small and unclear whether or not it is significant.

We have included statistical analysis in several graphs. As the referee anticipated, we found that the effect of cohesin is indeed not significant ($p > 0.05$) when considering individual minichromosomes (Figs 4B and 5C-D). Only the fact that the effect of cohesin was reproduced in distinct minichromosomes attained statistical significance (Figure 5E). It was appropriate to tone down the conclusions regarding cohesin, as the referee indicated.

Minor points:

1. The notion that SMC inactivation could lead to disruption in the equilibrium of top2 reactions has been proposed before, in the context of sister chromatid resolution (Sen et al 2016; Piskadlo et al 2017). This should be mentioned earlier in the introduction rather than simply in the discussion.

We agree. These references are now commented in the introduction.

2. Figures could be improved to have a more immediate read-out of the experimental layout. For example, in Fig. 2B, it would be good to have directly in the figure what 1, 2, 3 and 4 are; also, it would be good to have a graphical correspondence between the gels and the intensity profiles, either by colouring the respective numbers or by assigning the corresponding numbers to the profiles. The same applies to Fig. 2E, (clarify what 1 and 2 are and provide some sort of graphical correspondence between the gels and the intensity profiles).

We agree. Figure 2 was hard to follow. As suggested, we have included layouts of the experiments. We have clarified the correspondence of gel lanes with the intensity profiles, and improved the visual comparison among them.

3. The identity of the control plasmid used should be mentioned (it is solely referred as "8 Kb negatively supercoiled DNA plasmid"), even if only in figure legend or M&M.

We have identified YEp24 as the control plasmid in the results section and included it in Figure S1 of the appendix.

4. On Page 6, is the top2-d83 form expressed in levels identical to those of top2? Or, in other words, is the concentration/amount of each in the crude lysate similar?

Both proteins are expressed in a comparable amount as it can be seen in the PAGE gels that we have included in Figure 2D.

5. On page 7, correct Pkn=0.22 (I believe it must be 0.022).

Thanks. We corrected this typo.

6. For the cell cycle specific analysis, it would be advantageous to clarify that temperature shift was performed after the arrest somewhere in the figure, figure legend or results description. I had to read the M&M section to really understand the experimental layout.

We have described in the results section and the legend of Figure 3 how temperature shifts were performed on arrested cells.

7. Regarding the size dependence, the "small" minichromosome used in Fig, 4, 5 and 6 are not always consistent (e.g. YRp5 is only used in Fig. 5 and YRp4 is not used in this figure). Is there any reason for that?

We have clarified in the result section why YRp5 was used instead of YRp3 or YRp4 (the strain is TRP+ and so we could not use this auxotrophic marker).

8. Page 11 - drosophila should be corrected to Drosophila.

Done.

9. The paragraph on the "Role of condensin during interphase" should be rephrased to clarify particular differences between yeast and metazoans. For example, the authors state that condensin is bound on equal amounts in interphase vs mitosis in yeast. But in metazoans only condensin II is bound in interphase hence mitotic chromosomes have far larger amounts of condensins. Accordingly, the chromosome territories studies pointed out refer to condensin II.

We have rephrased this paragraph to indicate that two condensin complexes (condensin I and II) are present in metazoans, and that the interphase chromosome studies pointed out refer to condensin II.

10. The sentence "In fact, this scenario already occurs in sister chromatids during metaphase, in which condensin promotes SCI removal and so counteracts the effect of cohesin that favors SCI formation by maintaining sister chromatids in close proximity" (page 10) should be followed by the appropriate references.

We have rephrased and referenced this sentence as follows:

"This scenario might be analogous to that occurs in mitotic chromatin, where cohesin may favor SCI formation by maintaining sister chromatids in close proximity (Goloborodko et al., 2016b, Piskadlo et al.,

2017, Sen et al., 2016), whereas condensin might be performing continuous rounds of LE to enforce the removal of SCI."

11. The subsequent sentence states that condensin interacts with top2. A direct physical interaction has been proposed in Bhat et al, but several studies have failed to confirm this observation (see for example, Lavoie et al 2000; Bhalla et al 2002; Cuvier & Hirano 2003). There is indeed a lot of evidence that they occupy similar genomic loci (based on immunofluorescence and ChIP data) but the most accepted view, to my knowledge, is that this does not rely on a direct physical interaction. Additionally, whether condensin stimulates top2 activity has also remained controversial (see Charbin et al NAR 2014). This sentence should be rephrased accordingly.

We have rephrased the paragraph and commented the indicated references as follows:

"Based on immunofluorescence and ChIP data, topo II occupies similar genomic loci to condensin and cohesin, but their functional interplay remains unknown. Some studies suggested that condensin can physically interact with topo II and stimulate its activity (Bhat et al., 1996, D'Ambrosio et al., 2008a). Yet, other studies have failed to confirm a physical interaction (Bhalla et al., 2002, Cuvier & Hirano, 2003, Lavoie et al., 2002) or a stimulatory effect (Charbin et al., 2014)".

12. The findings that condensin depletion do not lead to changes in DNA supercoiling are indeed very interesting as they exclude a prior model to explain the functional interplay between condensin and top2 (proposed in Baxter et al). It would be interesting to refer to recent studies that also describe that condensin compacts DNA at the same rate, independently of the topological state of the DNA (Eeftens et al EMBO J 2017).

We have added and commented this study in the discussion about DNA supercoiling by condensin.

13. throughout the manuscript, but especially in the methods section, when abbreviating for degrees Celsius authors alternate between C, °C oC. One single form should be adopted and used in the entire manuscript.

We changed all to °C.

14. the description of yeast cell lysis in "Culture of yeast SMC mutants and DNA extraction" is good, as it allows for reproducibility. The same degree of detail should be used in "Topo II activity in crude yeast lysates"

We have described the procedure in more detail.

15. raffinose should be corrected to raffinose

Done.

16. Why were top ts mutants incubated at 30°C for the restrictive temperature instead of 35°C?

Done. We corrected this typo to 35 °C.

17. correct 50-ml to 50 ml

Done.

18. page 11 replace "transfected" by "transformed" (the most commonly used expression for yeast)

Done.

19. confirm K5832 genotype.

Done.

Referee #3

The authors should use graphics in the figures, eg. Figure S2, to explain the experimental results. For readers that are not experienced in this type of analyses it will be very difficult to appreciate the information, in particular on the linkage analysis figures. There is a great variation between these graphs in different figures, which is probably the nature of the experiment. However, it would be very helpful to see a "positive control" that shows which type of change can be expected in the figure when linkage is changed. Also, very specific terminology, eg. "thermal Lk distribution" or "precluding the oversimplification capacity of topo II" should be introduced or simplified. Also, the authors discuss several times "negative Lk values" but there is no quantitation of this somewhere in the manuscript. Further, the authors should thoroughly revise their manuscript and avoid overly complicated language as in this sentence: "*Our results show that disrupting the simplification activity of cellular topo II does not upturn the low DNA knotting probability of chromatin.*". Why not just say "... increase the DNA knotting ability ..". There are also a number of grammar problems, eventually proofreading by a native speaker could help.

We agree. Some terminology and figures were hard to follow. Therefore, in line also with reviewer #1 suggestions, we have done the following:

We included experimental layouts in Figure 2 and clarified the correspondence of gel lanes with intensity profiles. We included intensity profiles of the Lk distribution of the minichromosome to facilitate the comparison with the LK changes in the control plasmid. We added Expanded View Figure 1 to explain in more detail "equilibrium" and "simplified" Lk distributions. We replaced Figure S2 by two new figures (Figures EV3 and EV4), which illustrate the layout and interpretation of 2D gel electrophoresis of Lk topoisomers and of DNA knots. In EV Figure EV3, we clarify how negative Lk values are quantified and explain why the shape of Lk distributions can vary in different 2D gels. Finally, we revised the text to avoid complicated sentences and terminology. We asked a native speaker to proofread the manuscript.

Any documentation that the used yeast strains have the expected protein deletion under the experimental conditions, for example a western blot with suitable antibodies or functional assays, is missing.

We did not include functional analyses of the yeast strains since they have been broadly used and characterized in previous studies. However, as requested, we have included drop growth assays in Appendix Figure S2, which corroborate that the thermo-sensitive mutants used behave as expected.

In Figure 2B the authors want to demonstrate that the incubation with topo II has an effect on the Lk distribution and that this effect disappears upon addition of a topoisomerase inhibitor. Unfortunately, the effect cannot be appreciated in this figure. The distribution in lanes 2-4 looks nearly the same. If there is a significant change the authors should highlight this. The intensity plots should be better labelled and brought in context with the gel lanes. If the inhibitor treatment makes lane 4 more similar to lane 2, also these intensity plots should be compared.

We agree. As also indicated by reviewer #1, Figure 2B was hard to follow. Thus, we have better labelled and clarified the correspondence of gel lanes with new intensity profiles to facilitate their comparison. It is now more visual that, in absence of ICRF-193, topo II narrows the Lk distribution (lane 3), whereas the

the presence of ICRF-193 stops topo II from narrowing it (lane 4). This is why lane 2 (relaxed with topo I) and lane 4 (topo II with ICRF) are nearly the same.

In Figure 2 C the authors resolve the linking distribution in a 2D gel to establish the difference with inhibitor treatment. They state "Following the addition of ICRF-193, the Lk distribution of YEp13 was not significantly altered, in sharp contrast to that observed in the control plasmid." It is unclear to which observation or quantification in the blot the authors refer to. A visual impression of the graph seems to be insufficient in the eyes of this reviewer.

We have revised the text and figure 2C to clarify this point. We have included new intensity plots of the Lk distributions of YEp13 minichromosome (-/+ ICRF-193) in Figure 2C to facilitate their visual comparison with the Lk changes (-/+ ICRF-193) of the control plasmid in Figure 2B.

In the text the authors state: "Before adding ICRF-193, YEp13 presented a distribution of topoisomers of negative ΔLk values (Fig S2)", I cannot see this in this figure.

We agree this notion was missing. In the new EV Figure 3 (replacing Figure S2), we have now described the position and quantify the negative ΔLk values of YEp13.

Then the authors nick the DNA to analyze knots and state "Following the addition of ICRF-193, the knot probability of YEp13 was not significantly altered ($P_{kn} \approx 0.04$)." To state this the authors should perform a statistics test (also for all other experiments containing quantifications).

As also indicated by reviewer #1, we show the statistic analyses in the plots. These corroborated that P^{kn} of YEp13 was not significantly altered following the addition of ICRF-193.

Also in Figure 2F in the upper panel it is not clear which change is anticipated if the effect would be positive. As suggested already above, can the authors include a positive control?

If the effect would be positive, Lk distribution of the minichromosome with Top2- $\Delta 83$ would be wider than with TOP2. The positive control for this experiment is the naked plasmid (YEp24) added to the reaction (Figure 2E). To clarify this, we included intensity plots of the Lk distribution of the minichromosome in Figure 2F. These plots evidence no changes in the Lk of the minichromosome in contrast to those observed in the control plasmid in Figure 2E.

In Figures 4, 5 and 6 the authors investigate the effect of cohesin, condensin and smc5/6 inactivation on plasmids and minichromosomes of different size, containing also different functional elements. Is there evidence that cohesin, condensin and smc5/6 associate with these plasmids and could the number of proteins associated/ binding sites present account for the outcome of the experiments?

The fact that changes of P^{kn} are independent of the copy number and functional elements of the minichromosomes argue against a significant influence of stoichiometry or specific binding sites of condensin and cohesin, which is consistent with the abundance and broad genomic distribution of these complexes. Although the presence of these complexes might increase in large minichromosomes, this would not change the conclusions of the study. Yet, we cannot discard that the lack of effect of smc5/6 on P^{kn} could reflect the lower abundance of this complex. We thank the reviewer for rising this issue, which we included in the revised discussion -Distinct effects of condensin and cohesin- as follows:

"Lastly, the distinct effects of condensin and cohesin could result from unequal binding to minichromosomes. This possibility, however, seems less likely considering the comparable abundance and broad chromosomal distribution of both complexes in budding yeast (Glynn et al., 2004, Wang et al., 2005). Accordingly, the effects of condensin and cohesin on Pkn are accentuated with DNA length independently of the functional elements present in the minichromosomes. Only the lack of effects observed upon the inactivation of the Smc5/6 complex could be attributed to the lower abundance of this complex in comparison to cohesin and condensin (Aragon, 2018)"

The model in Figure 7 illustrates, how condensin could promote the removal of DNA knots. However, also the "loop extrusion" part of the model requires topo II. So, if the model is correct, one would have expected an effect of the topo II inactivation on the knotting efficiency in Fig. 2. An experiment using the double mutant should demonstrate whether this activity of condensin is indeed topo II dependent.

In previous studies (Valdes et al 2018), we already showed that topo II inactivation *per se* does not alter P^{kn} values since both DNA knotting and unknotting are inhibited. The knotting-unknotting balance becomes thereby frozen. According to our model, condensin inactivation changes this balance in a topo II dependent manner. Thus, we thank the reviewer for suggesting the experiment to validate this.

We constructed the double mutant (*top2-4 smc2-8*) as described in the methods section, and examined the P^{kn} of YEp13. The results are shown in a new figure (Expanded View Figure 5) and described in the results - Condensin inactivation boosts the occurrence of chromatin knots- (second paragraph) as follows:

"To verify that the smc2-8 allele was causing the three-fold increase of Pkn, we introduced this mutation in strains JCW25 (TOP2) and JCW26 (top2-4) (Trigueros & Roca, 2001). Upon shifting these cells to 35°C, the Pkn of YEp13 increased again about three-fold in the smc2-8 TOP2 cells (Fig EV5). However, DNA knot formation did not change in the smc2-8 top2-4 double mutant, which corroborated that topo II activity is required to produce the Pkn changes induced by condensin (Fig EV5)."

Minor suggestion:

In the introduction Fig.1, the authors mention also catenanes that might be present in G2 phase cells. Can these be observed in those assays?

We did not identify bands compatible with catenanes in our assays. To see catenated rings, we would have to adjust electrophoresis conditions and blot the upper part of the gels. These post-replication catenanes have been well characterized in previous studies (Baxter et al., 2011; Sen et al., 2016). We focused thereby in the presence of DNA knots, which better reflect the spontaneous entanglement of intracellular chromatin irrespective of DNA replication.

Thank you for submitting your revised manuscript for our consideration. It has now been seen again by the two previous referees (see comments below), and I am pleased to inform you that we have now accepted it for publication in The EMBO Journal.

Referee #1:

In this revised version of the manuscript, the authors have addressed most of my concerns and I am therefore happy to recommend its publication. The new focus of the results on the more convincing aspects of their work (regarding condensin) make this a much stronger paper. Also, the manuscript and results are now much easier to follow.

Referee #3:

Dyson and coworkers addressed the generation and resolution of DNA knots by topoisomerase II, condensin, cohesin and the smc5/6 complex using plasmids and minichromosomes of different sizes that were transfected into yeast cells.

They observe that inactivation of condensin increases the knots in a minichromosome and conclude that the loop extrusion activity of condensin might have a role in resolving DNA knots. This observation is very interesting and increases our understanding how cohesin and condensin organize chromatin during mitotic chromosome compaction and the role of condensin on interphase chromatin.

This reviewer had a number of concerns and criticisms in the first round of reviewing which were solved to my satisfaction. The manuscript was considerably improved by new data, reorganization of figures and also revision of the text.

There are no further concerns.

Corresponding Author Name: JOAQUIM ROCA

Journal Submitted to: The EMBO Journal

Manuscript Number: EMBOJ-2020-105393